# Slicing Mutual Information Generalization Bounds for Neural Networks

## Abstract

The ability of machine learning (ML) algorithms to generalize to unseen data has been studied through the lens of information theory, by bounding the generalization error with the input-output mutual information (MI), *i.e.* the MI between the training data and the learned hypothesis. These bounds have limited empirical use for modern ML applications (e.g., deep learning) since the evaluation of MI is difficult in high-dimensional settings. Motivated by recent reports of significant low-loss compressibility of neural networks, we study the generalization capacity of algorithms that *slice* the parameter space, *i.e.* train on a random lower-dimensional subspace. We derive information-theoretic bounds on generalization error in this regime and discuss an intriguing connection to the $k$-Sliced Mutual Information, an alternative measure of statistical dependence that scales well with dimension. We also propose a rate-distortion framework that allows generalization bounds to be obtained if the weights are simply *close to* the random subspace, and we propose a training procedure that exploits this flexibility. The computational and statistical benefits of our approach allow us to empirically estimate the input-output information of these neural networks and compute their information-theoretic generalization bounds, a task which was previously out of reach.

## 1 Introduction

Generalization is a fundamental aspect of machine learning, where models optimized to perform well on training data are expected to perform similarly well on test data drawn from the same underlying data distribution. Neural networks (NNs), in particular, are able to both achieve high performance on training data and generalize well to test data, allowing them to achieve excellent test performance on complex tasks. Despite this empirical success, however, the architectural factors influencing how well a neural network generalizes are not fully understood theoretically, motivating a substantial body of work using a variety of tools to bound their generalization error (Jiang et al., 2020b), e.g., PAC-Bayes (Dziugaite & Roy, 2017) and information theory (Xu & Raginsky, 2017).

We formally describe the generalization problem as follows. Let Z be the input data space (e.g. the set of feature-label pairs $z = (x, y)$), $\mu$ a probability distribution on Z, $W \subseteq \mathbb{R}^D$ the hypothesis space (e.g. weights of a NN), and $\ell : W \times Z \to \mathbb{R}_+$ a loss function (e.g. the classification error). The training procedure seeks to find a $w \in W$ with low *population risk* given by $\mathcal{R}(w) \triangleq \mathbb{E}_{Z \sim \mu}[\ell(w, Z)]$. In practice, computing $\mathcal{R}(w)$ is difficult since $\mu$ is generally unknown: one only observes a dataset comprising a finite number of samples from $\mu$. Instead, given a training dataset $S_n \triangleq \{z_i \in Z, i = 1, \ldots, n\}$, $(z_i)_{i=1}^n$ independently and identically distributed from $\mu$, we can measure the *empirical risk* $\widehat{\mathcal{R}}_n(w) \triangleq \frac{1}{n} \sum_{i=1}^n \ell(w, z_i)$. A learning algorithm can then be described as a function $\mathcal{A} : Z^n \to W$ which returns the optimal hypothesis $W$ learned from $S_n$. In general, $W$ is random, and we denote its probability distribution by $P_{W|S_n}$. The *generalization error* of $\mathcal{A}$ is then $\text{gen}(\mu, \mathcal{A}) \triangleq \mathbb{E}[\mathcal{R}(W) - \widehat{\mathcal{R}}_n(W)]$ where the expectation $\mathbb{E}$ is taken with respect to (w.r.t.) the joint distribution of $(W, S_n)$, *i.e.*, $P_{W|S_n} \otimes \mu^{\otimes n}$. The higher $\text{gen}(\mu, \mathcal{A})$, the more $\mathcal{A}$ overfits when trained on $S_n \sim \mu^{\otimes n}$.

**Information-theoretic bounds.** In recent years, there has been a flurry of interest in using theoretical approaches to bound $\text{gen}(\mu, \mathcal{A})$ using *mutual information* (MI). The MI between two random variables $X$ and $Y$ is defined as $\mathsf{I}(X; Y) = \iint p(x, y) \log \left( \frac{p(x,y)}{p(x)p(y)} \right) \mathrm{d}x \, \mathrm{d}y$, where $p(x, y)$ denotes

the joint distribution of $(X, Y)$ at $(x, y)$, and $p(x), p(y)$ are the marginals. The most common information-theoretic bound on generalization error was introduced by Xu & Raginsky (2017) and depends on $\mathsf{I}(W; S_n)$, where $W$ is the optimal hypothesis learned from $S_n$. We recall the formal statement below.

**Theorem 1.1** (Xu & Raginsky, 2017). *Assume that $\ell(w, Z)$ is $\sigma$-sub-Gaussian[1] under $Z \sim \mu$ for all $w \in \mathrm{W}$. Then, $|\mathrm{gen}(\mu, \mathcal{A})| \leq \sqrt{2\sigma^2 \, \mathsf{I}(W; S_n)/n}$, where $W = \mathcal{A}(S_n)$.*

Examples of $\sigma$-sub-Gaussian losses include $\ell(w, Z) \sim \mathcal{N}(0, \tau^2)$ (in that case, $\sigma = \tau$) and $\ell(w, Z) \leq C$ (by Hoeffding's lemma, $\sigma = C/2$). Subsequently, Bu et al. (2019) used the averaging structure of the empirical loss to derive a bound that depends on $\mathsf{I}(W; Z_i)$, $i \in \{1, \ldots, n\}$. By evaluating MI on *individual* data points $Z_i$, rather than the entire training dataset $S_n$, one can obtain a tighter bound than Xu & Raginsky (2017) in certain problems (Bu et al., 2019, §IV).

**Theorem 1.2** (Bu et al., 2019). *Assume that $\ell(\tilde{W}, \tilde{Z})$ is $\sigma$-sub-Gaussian under $(\tilde{W}, \tilde{Z}) \sim P_W \otimes \mu$. Then, $|\mathrm{gen}(\mu, \mathcal{A})| \leq (1/n) \sum_{i=1}^{n} \sqrt{2\sigma^2 \, \mathsf{I}(W; Z_i)}$, where $W = \mathcal{A}(S_n)$.*

Most information-theoretic bounds, however, suffer from the fact that the dimension of $W$ can be large when using modern ML models, e.g. NNs. Indeed, the sample complexity of MI estimation scales poorly with dimension (Paninski, 2003). Collecting more samples of $(W, Z_i)$ can be expensive, especially with NNs, as one realization of $W \sim P_{W|S_n}$ requires one complete training run. Moreover, McAllester & Stratos (2020) recently proved that estimating MI from finite data have important statistical limitations when the underlying MI is large, e.g. hundreds of bits.

**Sliced neural networks.** While modern neural networks use large numbers of parameters, common architectures can be highly compressible by *random slicing*: Li et al. (2018) found that restricting $W \in \mathbb{R}^D$ during training to lie in a $d$-dimensional subspace spanned by a random matrix (with $d \ll D$) not only provides computational advantages, but does not meaningfully damage the performance of the neural network, for appropriate choice of $d$ (often two orders of magnitude smaller than $D$). They interpreted this fact as indicating *compressibility* of the neural network architecture up to some *intrinsic dimension* $d$, below which performance degrades. This framework has recently been applied by Lotfi et al. (2022) to significantly improve PAC-Bayes generalization bounds, to the point where they closely match empirically observed generalization error.

**Sliced mutual information.** It is a natural question whether we can leverage the compression created by slicing to obtain tighter and computationally-friendly information-theoretic generalization bounds. Intriguingly, a parallel line of work has considered slicing mutual information itself, yielding significant sample complexity and computational advantages in high-dimensional regimes. Goldfeld & Greenewald (2021) and Goldfeld et al. (2022) slice the arguments of MI via random $k$-dimensional projections, thus defining the $k$-*Sliced Mutual Information* (SMI) between $X \in \mathbb{R}^{d_x}$ and $Y \in \mathbb{R}^{d_y}$ as

$$\mathsf{SI}_k(X; Y) = \iint \mathsf{I}^{A,B}(\mathrm{A}^\top X; \mathrm{B}^\top Y) \, \mathrm{d}(\sigma_{k,d_x} \otimes \sigma_{k,d_y})(\mathrm{A}, \mathrm{B}) \,,$$

where $\mathsf{I}^{A,B}(\mathrm{A}^\top X; \mathrm{B}^\top Y)$ is the *disintegrated MI* between $\mathrm{A}^\top X$ and $\mathrm{B}^\top Y$ given $(A, B)$ (Negrea et al., 2019, Definition 1.1) and $\sigma_{k,d}$ is the Haar measure on $\mathrm{St}(k, d)$, the Stiefel manifold of $d \times k$ matrices with orthonormal columns. $\mathsf{SI}_k$ has been shown to retain many important properties of MI (Goldfeld et al., 2022), and—more importantly—the statistical convergence rate for estimating $\mathsf{SI}_k(X; Y)$ depends on $k$ but not the ambient dimensions $d_x, d_y$. This provides significant advantages over MI, whose computation generally requires an exponential number of samples in $\max(d_x, d_y)$ (Paninski, 2003). Similar convergence rates can be achieved while slicing in only one dimension, e.g. $X$, if samples from the conditional distribution of $X|Y = y$ are available (Goldfeld & Greenewald, 2021), yielding

$$\mathsf{SI}_k^{(1)}(X; Y) = \int_{\mathrm{St}(k, d_x)} \mathsf{I}^A(\mathrm{A}^\top X; Y) \, \mathrm{d}\sigma_{k,d_x}(\mathrm{A}) \,. \tag{1}$$

Recently, Wongso et al. (2023) empirically connected generalization to $\mathsf{SI}_k^{(1)}(T; Y)$ between the true class labels $Y$ and the hidden representations $T$ of NNs.

---

[1]A random variable $X$ is $\sigma$-sub-Gaussian ($\sigma > 0$) under $\mu$ if for $t \in \mathbb{R}$, $\mathbb{E}_\mu[e^{t(X - \mathbb{E}_\mu[X])}] \leq e^{\sigma^2 t^2/2}$.

**Our contributions.** Motivated by the above, we introduce information-theoretic bounds studying the generalization capacity of learning algorithms trained on random subspaces. Our bounds demonstrate that neural networks that are "compressible" via random slicing have significantly better information-theoretic generalization guarantees. We also find an intriguing connection to SMI, which we explore in learning problems where the information-theoretic generalization bounds are analytically computable. We then leverage the computational and statistical benefits of our sliced approach to empirically compute nonvacuous information-theoretic generalization bounds for various neural networks.

We further increase the practicality of our approach by using the *rate-distortion* based framework introduced by Sefidgaran et al. (2022) to extend our bounds to the setting where the weight vector $W$ only approximately lies on random subspace. This extension applies when the loss is Lipschitz w.r.t. the weights, which we promote using techniques from Bethune et al. (2023). As Sefidgaran et al. (2022) did for quantization, this allows us to apply generalization bounds based on projection and quantization to networks whose weights are unrestricted. We tighten the bound by using regularization in training to encourage the weights to be close to the random subspace. We find this regularization not only improves the generalization bound, but also test performance.

## 2 RELATED WORK

**Compression of neural networks.** Our work focuses on random projection and quantization (c.f. Hubara et al. (2016)) as tools for compressing neural networks. Many other compression approaches exist, however Cheng et al. (2017); Hutson (2020), e.g. pruning Dong et al. (2017); Blalock et al. (2020), low rank compression Wen et al. (2017), and optimizing architectures via neural architecture search and metalearning Pham et al. (2018); Cai et al. (2020); Finn et al. (2017). Further exploring alternative compression approaches from an information-theoretic generalization bound perspective is an interesting avenue for future work.

**Compressibility and generalization.** A body of work has emerged using various notions of compressibility of neural networks to obtain improved generalization bounds, for instance Arora et al. (2018); Hsu et al. (2021); Kuhn et al. (2021); Sefidgaran et al. (2022), and *fractal dimension* based on the intrinsic dimension of the optimization dynamics, e.g. Simsekli et al. (2020).

**Conditional MI generalization bounds.** Following Xu & Raginsky (2017) and Bu et al. (2019), which treat the training data as random, Steinke & Zakynthinou (2020) instead obtain a bound where the dataset is fixed (*i.e. conditioned* on a dataset). This framework assumes that two independent datasets are available, and random Bernoulli indicator variables create a random training set by randomly selecting which of the two datasets to use for the $i$th training point. This approach has the advantage of creating a generalization bound involving the mutual information between the learned weights and a set of *discrete* random variables, in which case the mutual information is always finite. Connections to other generalization bound strategies and to data privacy are established by Steinke & Zakynthinou (2020). Followup works tightened these bounds by considering the conditional mutual information between the indicator variables and either the *predictions* (Harutyunyan et al., 2021; Haghifam et al., 2022) or *loss* (Wang & Mao, 2023) of the learned model rather than the weights. A practical limitation of this general approach is that it requires a second dataset (or *supersample*) to compute the conditional mutual information, whereas this extra data could be used to get a better estimate of the test error (hence, the generalization error) directly. Additionally, some of these bounds depend on a mutual information term between low-dimensional variables (e.g., functional CMI-based bounds (Harutyunyan et al., 2021)), which can be evaluated efficiently but does not inform practitioners for selecting model architectures. Exploring slicing in the context of the conditional MI framework is beyond the scope of our paper, but is a promising direction for future work.

**Other generalization bounds for neural networks.** Beyond the information-theoretic frameworks above, many methods bound the generalization of neural networks. Classic approaches in learning theory bound generalization error with complexity of the hypothesis class (Bartlett & Mendelson, 2002; Vapnik & Chervonenkis, 2015), but these fail to explain the generalization ability of deep neural networks with corrupted labels (Zhang et al., 2017). More successful approaches include the PAC-Bayes framework (including Lotfi et al.'s work above, whose use of slicing inspired our work), margin-based approaches (Koltchinskii et al., 2002; Kuznetsov et al., 2015; Chuang et al., 2021),

and even empirically-trained prediction not based on theoretical guarantees (Jiang et al., 2020a; Lassance et al., 2020; Natekar & Sharma, 2020; Schiff et al., 2021). Each approach has its own benefits and drawbacks; for instance, many of the tightest predictions are highly data-driven and as a result may provide limited insight into the underlying sources of generalization and how to design networks to promote it.

**Our work.** In the context of the above literature, the purpose of this work is to use slicing to dramatically improve the tightness of *input-output information-theoretic generalization bounds* for neural networks. We achieve nonvacuous bounds for NNs of practical size, which to our knowledge have not been seen using Theorems 1.1 and 1.2 above. That said, our bounds (unsurprisingly) are still looser than generalization bounds available through some other techniques mentioned above, particularly those employing additional data (e.g. data-driven PAC-Bayes priors (Lotfi et al., 2022) or the super-sample of conditional MI bounds (Wang & Mao, 2023)) or involving some kind of trained or ad hoc prediction function. Regardless, continuing to improve information-theoretic bounds is a fruitful endeavor that serves to better understand the connection between machine learning and information theory, and to gain insights that can drive algorithmic and architectural innovation. For instance, our rate-distortion bounds informed our creation of a regularization technique for NNs, which not only yields generalization *bounds* but also improves generalization *performance*.

## 3 SLICED INFORMATION-THEORETIC GENERALIZATION BOUNDS

We establish information-theoretic generalization bounds for any algorithm $\mathcal{A}^{(d)}$ that samples a random projection matrix $\Theta \sim P_\Theta$ of size $D \times d$ with $d < D$ and $\Theta^\top \Theta = \boldsymbol{I}_d$ and returns a trained model with parameters that lie on $\mathrm{W}_{\Theta,d} \triangleq \{w \in \mathbb{R}^D : \exists w' \in \mathbb{R}^d \text{ s.t. } w = \Theta w'\}$. In other words, $\mathcal{A}^{(d)}$ is a *slicing* algorithm that restricts the weights of a neural network to a random $d$-dimensional subspace $\Theta$. Generally speaking, the training procedure will boil down to optimizing the subspace coefficients $w' \in \mathbb{R}^d$ given $\Theta$, and $P_\Theta$ is e.g. the uniform distribution on the Stiefel manifold $\mathrm{St}(d, D)$.

We analyze the generalization error of models trained by $\mathcal{A}^{(d)}$. In this setting, the population risk and empirical risk are respectively defined as

$$\mathcal{R}^\Theta(w') \triangleq \mathbb{E}_{Z \sim \mu}[\ell^\Theta(w', Z)] \quad \text{and} \quad \widehat{\mathcal{R}}_n^\Theta(w') \triangleq \frac{1}{n} \sum_{i=1}^n \ell^\Theta(w', z_i), \quad \forall w = \Theta w' \in \mathrm{W}_{\Theta,d}, \quad (2)$$

and $\ell^\Theta(w', z) \triangleq \ell(\Theta w', z)$. The generalization error of $\mathcal{A}^{(d)}$ is $\mathrm{gen}(\mu, \mathcal{A}^{(d)}) = \mathbb{E}[\mathcal{R}^\Theta(W') - \widehat{\mathcal{R}}_n^\Theta(W')]$ with the expectation taken over $P_{W'|\Theta,S_n} \otimes P_\Theta \otimes \mu^{\otimes n}$. Here, we take the expectation with respect to $\Theta$ to obtain a number that does not depend on $\Theta$.[2]

### 3.1 BOUNDING GENERALIZATION ERROR VIA $\mathsf{I}^\Theta(W'; S_n)$ OR $\mathsf{I}^\Theta(W'; Z_i)$

The disintegrated mutual information between $X$ and $Y$ given $U$ is defined as

$$\mathsf{I}^U(X; Y) = \iint p(x, y|u) \log \left( \frac{p(x, y|u)}{p(x|u)p(y|u)} \right) \mathrm{d}x \, \mathrm{d}y, \quad (3)$$

where $p(x, y|u)$ denotes the conditional distribution of $(X, Y)$ at $(x, y)$ given $U = u$, and $p(x|u)$ (resp., $p(y|u)$) is the conditional distribution of $X$ at $x$ (resp., $Y$ at $y$) given $U = u$.

**Theorem 3.1.** *Assume $\ell^\Theta(w', Z)$ is $\sigma$-sub-Gaussian under $Z \sim \mu$ for all $w' \in \mathbb{R}^d$ and $\Theta \in \mathbb{R}^{D \times d}$, $\Theta^\top \Theta = \boldsymbol{I}_d$. Then,*

$$|\mathrm{gen}(\mu, \mathcal{A}^{(d)})| \leq \sqrt{\frac{2\sigma^2}{n}} \, \mathbb{E}_{P_\Theta} \left[ \sqrt{\mathsf{I}^\Theta(W'; S_n)} \right]. \quad (4)$$

While state-of-the-art MI-based bounds depend on $\mathsf{I}(W; S_n)$ (e.g., Xu & Raginsky, 2017), we leverage the constraint set $\mathrm{W}_{\Theta,d}$ to construct a bound in terms of $\mathsf{I}^\Theta(W'; S_n)$. Since $W'$ is lower-dimensional, our bound can be estimated more easily in practice. Besides, due to the compression

---

[2]In practice, for bounding the generalization of a specific model, it is often sufficient to simply fix $\Theta$ to be whatever was sampled and used by $\mathcal{A}^{(d)}$ to obtain the model being used in practice.

of the hypothesis space, we will see our bound (4) is tighter in practice than that of Xu & Raginsky (2017), which ignores the intrinsic dimension $d < D$ of the hypothesis space.[3] This approach, also referred to as *disintegration*, has been used to tighten MI-based generalization bounds (Hellström et al., 2023, §4.3): Bu et al. (2019) used disintegration to derive bounds in terms of individual-sample MI, $\mathsf{I}(W; Z_i)$, which are tighter than the full-sample counterpart of Xu & Raginsky (2017). To the best of our knowledge, however, we provide the first bounds where disintegration is applied to account for the intrinsic dimension of the hypothesis space.

**Discrete hypothesis space and dependence on $d$.** Using analogous arguments as Xu & Raginsky (2017, §4.1), we can upper-bound $\mathbb{E}_{P_\Theta}\left[\sqrt{\mathsf{I}^\Theta(W'; S_n)}\right]$ when $W'$ is a discrete random variable. Indeed, for a fixed $\Theta$, if $W'$ given $\Theta$ takes $K$ possible values, then $\mathsf{I}^\Theta(W'; S_n) \leq H^\Theta(W') \leq \log(K)$, where $H^\Theta(W')$ is the entropy of $W'$ conditioned on $\Theta$. In that case, and under the assumptions of Theorem 3.1, $|\text{gen}(\mu, \mathcal{A}^{(d)})| \leq \sqrt{2\sigma^2 \log(K)/n}$. Consider the setting where the discrete hypothesis space is a quantization of each element in $W'$ into $B$ bins, then, $K = B^d$ and $|\text{gen}(\mu, \mathcal{A}^{(d)})| \leq \sqrt{2\sigma^2 d \log(B)/n}$. This bound rapidly decreases as $d$ shrinks, showing the benefit of keeping $d$ small as we propose. On the other hand, decreasing $d$ may *increase* the training error, implying a tradeoff between generalization error and training error when selecting $d$.

Next, we adapt the strategy of Bu et al. (2019) and construct a bound in terms of individual sample-based MI $\mathsf{I}^\Theta(W'; Z_i)$, instead of $\mathsf{I}^\Theta(W'; S_n)$.

**Theorem 3.2.** *Assume that $\ell^\Theta(\tilde{W}', \tilde{Z})$ is $\sigma_\Theta$-sub-Gaussian under $(\tilde{W}', \tilde{Z}) \sim P_{W'|\Theta} \otimes \mu$ for all $\Theta \in \mathbb{R}^{D \times d}$, $\Theta^\top \Theta = \boldsymbol{I}_d$, where $\sigma_\Theta$ is a positive constant that may depend on $\Theta$. Then,*

$$|\text{gen}(\mu, \mathcal{A}^{(d)})| \leq \frac{1}{n} \sum_{i=1}^n \mathbb{E}_{P_\Theta}\left[\sqrt{2\sigma_\Theta^2 \mathsf{I}^\Theta(W'; Z_i)}\right]. \tag{5}$$

**Discussion.** The bound in Theorem 3.1 may be vacuous in certain settings. For instance, if $W' = g(S_n)$ where $g$ is a smooth, non-constant and deterministic function (that may depend on $\Theta$), then $\mathsf{I}^\Theta(W'; S_n) = +\infty$, as in the Gaussian mean estimation problem studied in Section 3.2. The bound in Theorem 3.2 overcomes this issue, as it depends on individual sample-based MI. In addition, Theorem 3.2 is a particular case of a more general result, where the sub-Gaussian condition is replaced by milder assumptions on the *cumulant-generating function* (CGF) of $\ell^\Theta(\tilde{W}', \tilde{Z})$ (given a fixed $\Theta$) defined for $t \in \mathbb{R}$ as $K_{\ell^\Theta(\tilde{W}', \tilde{Z})}(t) = \log \mathbb{E}[e^{t(\ell^\Theta(\tilde{W}', \tilde{Z}) - \mathbb{E}[\ell^\Theta(\tilde{W}', \tilde{Z})])}]$, with expectations over $P_{W'|\Theta} \otimes \mu$. Due to space limit, we give the formal statement of our general result in Appendix A.1 (see Theorem A.2). We emphasize that Theorem A.2 has a broader applicability than Theorems 3.1 and 3.2, which is useful to bound generalization errors based on non-sub-Gaussian losses, as in the linear regression problem considered next.

## 3.2 APPLICATIONS AND CONNECTION TO THE SLICED MUTUAL INFORMATION

To further illustrate the advantages of our bounds as compared to those of Bu et al. (2019), we apply them to two specific settings. These examples also allow us to draw a connection with $k$-SMI.

**Gaussian mean estimation.** Denote by $\|\cdot\|$ the Euclidean norm and $\boldsymbol{0}_D$ the $D$-dimensional zero vector. We consider the problem of estimating the mean of $Z \sim \mathcal{N}(\boldsymbol{0}_D, \boldsymbol{I}_D)$ via empirical risk minimization. The training dataset $S_n = (Z_1, \ldots, Z_n)$ consists of $n$ i.i.d. samples from $\mathcal{N}(\boldsymbol{0}_D, \boldsymbol{I}_D)$. Our objective is $\min_{w \in \mathsf{W}_{\Theta, d}} \widehat{\mathcal{R}}_n(w) \triangleq \frac{1}{n} \sum_{i=1}^n \|w - Z_i\|^2$. We prove in Appendix A.3 that *(i)* $W' = \Theta^\top \bar{Z}$, $\bar{Z} \triangleq (1/n) \sum_{i=1}^n Z_i$, *(ii)* $\text{gen}(\mu, \mathcal{A}^{(d)}) = 2\sigma^2 d/n$, and *(iii)* $\ell^\Theta(\tilde{W}', \tilde{Z}) = \|\Theta \tilde{W}' - \tilde{Z}\|^2$ is sub-Gaussian under $(\tilde{W}', \tilde{Z}) \sim P_{W'|\Theta} \otimes \mu$ for all $\Theta$. By applying Theorem 3.2,

$$\text{gen}(\mu, \mathcal{A}^{(d)}) \leq \frac{2}{n} \sqrt{d\left(1 + \frac{1}{n}\right)^2 + (D - d) \sum_{i=1}^n \mathbb{E}_{P_\Theta}\left[\sqrt{\mathsf{I}^\Theta(\Theta^\top \bar{Z}; Z_i)}\right]}. \tag{6}$$

---

[3]By the data processing inequality, our bound can be shown theoretically to be tighter for mean estimation and linear regression, discussed below.

The connection between $\mathrm{gen}(\mu, \mathcal{A}^{(d)})$ and SMI results from using Jensen's inequality on (6) and the fact that $W' = \Theta^\top W$, where $W \triangleq \arg\min_{w \in \mathbb{R}^D} \widehat{\mathcal{R}}_n(w) = \bar{Z}$ is the solution of the unconstrained problem. Indeed, this shows $\mathbb{E}_{P_\Theta}[\sqrt{\mathsf{I}^\Theta(\Theta^\top \bar{Z}; Z_i)}] < \sqrt{\mathsf{SI}_d^{(1)}(W; Z_i)}$, $i \in \{1, \ldots, n\}$. Note that by the data processing inequality, $\mathsf{I}^\Theta(W'; Z_i) \leq \mathsf{I}^\Theta(W; Z_i)$, thus we obtain tighter bounds as compared to strategies which ignore the existence of an intrinsic dimension. In the limit case $d = D$, our bound (6) boils down to the one in Bu et al. (2019), since $\mathsf{SI}_D^{(1)}(W; Z_i) = \mathsf{I}(W; Z_i)$. When $d < D$, the bound by Bu et al. (2019) is not applicable: the CGF of $\ell(\Theta \tilde{W}', \tilde{Z})$ for $(\Theta \tilde{W}', \tilde{Z}) \sim P_{\Theta W'} \otimes \mu$ cannot be derived analytically since $P_{\Theta W'}$ is not Gaussian (as opposed to the case $d = D$). In contrast, our bound depends on the CGF of $\ell^\Theta(\tilde{W}', \tilde{Z})$ for $(\tilde{W}', \tilde{Z}) \sim P_{W'|\Theta} \otimes \mu$, which is available in closed form thanks to conditioning on $\Theta$. Finally, analogously to the unconstrained problem (Bu et al., 2019, §IV.A), our bound (6) is suboptimal since it is in $\mathcal{O}(1/\sqrt{n})$ as $n \to +\infty$, while the true generalization error is $2d/n$. We provide the derivations in Appendix A.3 and illustrate our bound vs. (Bu et al., 2019) in Figure 1.

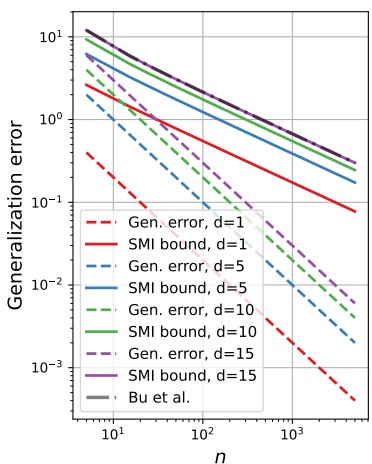

Figure 1: Evaluation of the mean estimation generalization bound of (6) for $D = 15$ in log-log scale, showing tighter generalization error bounds as $d$ decreases. Note the bound in Bu et al. (2019) only applies for $d = D$.

**Linear regression.** Consider $n$ i.i.d. samples $(x_1, \ldots, x_n)$, $x_i \in \mathbb{R}^D$ and a response variable $y = (y_1, \ldots, y_n)$, $y_i \in \mathbb{R}$. The goal of $\mathcal{A}^{(d)}$ is $\min_{w \in \mathrm{W}_{\Theta, d}} \widehat{\mathcal{R}}_n(w) \triangleq (1/n)\|y - Xw\|^2$, where $X \in \mathbb{R}^{n \times D}$ denotes the design matrix. We show that if $n \geq D$, then $W' = (\Theta X^\top X \Theta^\top)^{-1} \Theta X^\top y$. Moreover, assume that $X$ is deterministic and $y_i = x_i^\top W^\star + \varepsilon_i$ where $W^\star \in \mathbb{R}^D$ and $(\varepsilon_i)_{i=1}^n$ i.i.d. from $\mathcal{N}(0, \sigma^2)$. Then, by applying Theorem A.2, we bound $\mathrm{gen}(\mu, \mathcal{A}^{(d)})$ by a function of $\mathsf{I}(\phi(\Theta, X)W; y_i)$, where $\phi(\Theta, X) \triangleq (\Theta X^\top X \Theta^\top)^{-1} \Theta (X^\top X)$ and $W \triangleq \arg\min_{w \in \mathbb{R}^D} \widehat{\mathcal{R}}_n(w)$, which can be interpreted as a generalized SMI with a non-isotropic slicing distribution that depends on the fixed $X$. The corresponding derivations are detailed in Appendix A.4.

## 3.3 RATE-DISTORTION GENERALIZATION BOUNDS

The above bounds require the learned weights $W$ to lie in $\mathrm{W}_{\Theta, d}$. When $d$ is very small, this constraint can be restrictive and significantly deteriorate the performance of the model, as we illustrate in Section 4. Besides, since our MI-based bounds generally increase with increasing $d$, it is important to keep $d$ small. Motivated by recent work applying rate-distortion theory to input-output MI generalization bounds (Sefidgaran et al., 2022), we establish the following result for *approximately* compressible weights and Lipschitz loss.

**Theorem 3.3.** *Consider* $\mathcal{A} : \mathrm{Z}^n \to \mathrm{W}$ *s.t.* $\mathcal{A}$ *may take* $\Theta \sim P_\Theta$ *into account to output $W$. Assume there exists $C > 0$ s.t.* $\ell(\tilde{W}, \tilde{Z}) \leq C$ *almost surely. Assume for any $z \in \mathrm{Z}$,* $\ell(\cdot, z) : \mathrm{W} \to \mathbb{R}_+$ *is $L$-Lipschitz, i.e.* $\forall (w_1, w_2) \in \mathrm{W} \times \mathrm{W}, |\ell(w_1, z) - \ell(w_2, z)| \leq L\|w_1 - w_2\|$. *Then,*

$$|\mathrm{gen}(\mu, \mathcal{A})| \leq 2L\mathbb{E}_{P_{W|\Theta} \otimes P_\Theta}\left[\|W - \Theta\Theta^\top W\|\right] + \frac{C}{n}\sum_{i=1}^n \mathbb{E}_{P_\Theta}\left[\sqrt{\frac{\mathsf{I}^\Theta(\Theta^\top W; Z_i)}{2}}\right]. \quad (7)$$

The proof of Theorem 3.3 consists in considering two models $\mathcal{A} : \mathrm{Z}^n \to \mathbb{R}^D$ and $\mathcal{A}' : \mathrm{Z}^n \to \mathrm{W}_{\Theta, d}$ such that $\mathcal{A}(S_n) = W$ may depend on $\Theta \sim P_\Theta$, and $\mathcal{A}'(S_n) = \Theta(\Theta^\top W)$; then using the triangle inequality to obtain $|\mathrm{gen}(\mu, \mathcal{A})| \leq |\mathrm{gen}(\mu, \mathcal{A}) - \mathrm{gen}(\mu, \mathcal{A}')| + |\mathrm{gen}(\mu, \mathcal{A}')|$; and finally bounding the first term (the *distortion* term) using the Lipschitz condition and the second term (the *rate* term) using our Theorem 3.2. Using similar arguments and applying Theorem 3.1, we derive another rate-distortion bound based on quantization, which does not require estimation of MI.

**Theorem 3.4.** *Assume the conditions of Theorem 3.3 hold. Furthermore, suppose that* $\|\Theta^\top W\| \leq M$ *for* $(W, \Theta) \sim P_{W|\Theta} \otimes P_\Theta$. *Consider a function $\mathcal{Q}$ quantizing $\Theta^\top W$ such that* $\|\Theta^\top W -$

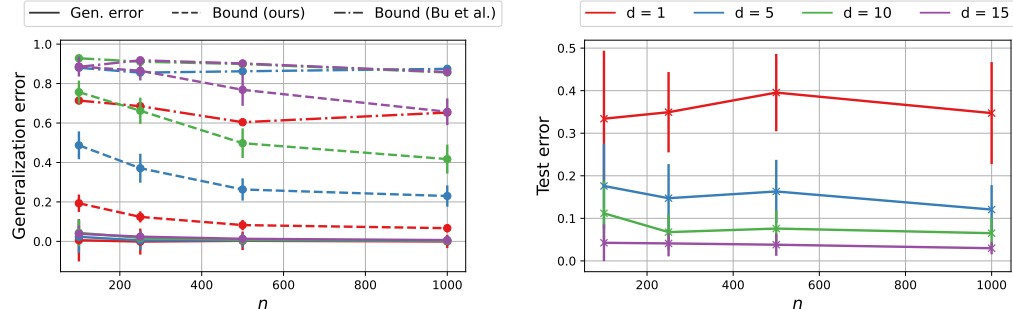

Figure 2: Illustration of our bound (5) and Bu et al. (2019) on binary classification of Gaussian data of dimension 20 with logistic regression trained on $W_{\Theta,d}$

$\mathcal{Q}(\Theta^\top W)\| \leq \delta$. *Then,*

$$|\text{gen}(\mu, \mathcal{A})| \leq 2L\mathbb{E}_{P_{W|\Theta}\otimes P_\Theta}\left[\|W - \Theta\mathcal{Q}(\Theta^\top W)\|\right] + C\mathbb{E}_{P_\Theta}\left[\sqrt{\frac{I^\Theta(\mathcal{Q}(\Theta^\top W); S_n)}{2n}}\right] \qquad (8)$$

$$\leq 2L\left(\mathbb{E}_{P_{W|\Theta}\otimes P_\Theta}\left[\|W - \Theta\Theta^\top W\|\right] + \delta\right) + C\sqrt{\frac{d\log(2M\sqrt{d}/\delta)}{2n}} . \qquad (9)$$

Note that $\|\Theta^\top W\| \leq M$ is a mild assumption, since in general, this is a result of enforcing Lipschitz continuity (e.g. the Lipschitz neural networks studied by Bethune et al. (2023) require weights with bounded norms). By setting $\delta = 1/\sqrt{n}$, our bound (9) decreases as $n \to +\infty$, which reflects the fact that training on more samples improves generalization. Besides, as $d$ goes to $D$, (9) converges to the bound of Xu & Raginsky (2017) and hence becomes vacuous for over-parameterized models where $D > n$ (e.g. NNs). Theorem 3.4 thus proves that accounting for compressibility of the hypothesis space can help improve existing generalization bounds that ignore this information.

Our theoretical findings provide concrete guidelines on how to tighten the generalization error bounds in practice. First, the value of the Lipschitz constant $L$ can be directly controlled through the design of the neural network, as we explain in Section 4. The term $\mathbb{E}_{P_{W|\Theta}\otimes P_\Theta}\|W - \Theta\Theta^\top W\|$ is controlled by adding it as a regularizer to the training objective, specifically, we add the regularization term $\lambda\mathbb{E}_{P_{W|\Theta}\otimes P_\Theta}\|W - \Theta\Theta^\top W\|$. This regularizer has the effect of encouraging solutions to be close to the subspace $W_{\Theta,d}$, *i.e.* having low distortion from the compressed weights. The choice of $d$ is also important and can be tuned to balance the MI term with the distortion required (how small $\lambda$ needs to be) to achieve low training error.[4]

## 4 EMPIRICAL ANALYSIS

To illustrate our findings and their impact in practice, we train several neural networks for classification, and evaluate their generalization error and our bounds. This requires compressing NNs (via random slicing and quantization) and estimating MI. We explain our methodology below, and refer to Appendix B.1 for more details and additional results. All our experiments can be reproduced with the source code provided in the supplementary material.

**Random projections.** To sample $\Theta \in \mathbb{R}^{D\times d}$ such that $\Theta^\top\Theta = I_d$, we construct an orthonormal basis using the singular value decomposition of a random matrix $\Gamma \in \mathbb{R}^{D\times d}$ whose entries are i.i.d. from $\mathcal{N}(0, 1)$. Since the produced matrix $\Theta$ is dense, the projection $\Theta^\top w$ induces a runtime of $\mathcal{O}(dD)$. To improve scalability, we use the sparse projector of Li et al. (2018) and the Kronecker product projector of Lotfi et al. (2022), which compute $\Theta^\top w$ in $\mathcal{O}(d\sqrt{D})$ and $\mathcal{O}(\sqrt{dD})$ operations respectively, and require storing only $\mathcal{O}(d\sqrt{D})$ and $\mathcal{O}(\sqrt{dD})$ matrix elements respectively.

**Quantization.** We use the quantizer of Lotfi et al. (2022), which simultaneously learns the quantized weights $W'$ and quantized levels $(c_1, \cdots, c_L)$. This allows us to highly compress NNs, and bypass

---

[4]Increasing $\lambda$ increases the weight on the regularization, effectively reducing the importance of empirical risk. Hence, empirical risk may rise, which in most cases will necessarily increase training error.

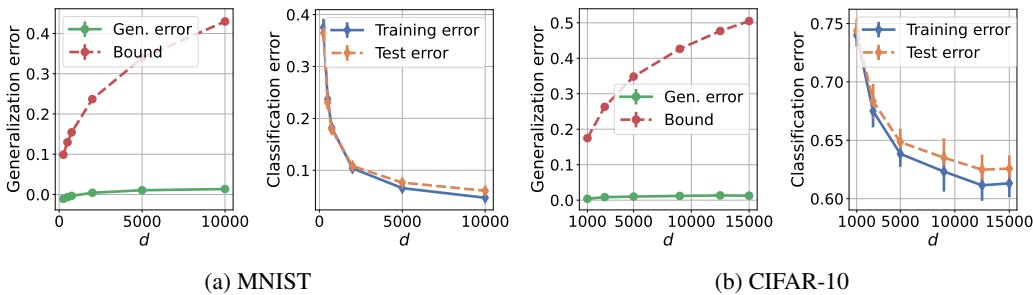

(a) MNIST

(b) CIFAR-10

Figure 3: Illustration of our generalization bounds with NNs for image classification. The weights are projected and quantized.

the estimation of MI: for $\Theta \sim P_\Theta$, $\mathsf{I}^\Theta(W'; S_n) \leq H(W'|\Theta) \leq \lceil d \times H(p) \rceil + L \times (16 + \lceil \log_2 d \rceil) + 2$ where $H(p) = -\sum_{k=1}^{L} p_k \log(p_k)$ and $p_k$ is the empirical probability of $c_k$.

**Estimating MI.** In our practical settings, the MI terms arising in the generalization bounds cannot be computed exactly, so we resort to two popular estimators: the $k$-nearest neighbor estimator ($k$NN-MI, Kraskov et al., 2004) and MINE (Belghazi et al., 2018). We obtain NaN values with $k$NN-MI for $d > 2$ thus only report the bounds estimated with MINE.

### 4.1 ILLUSTRATION OF GENERALIZATION BOUNDS FOR MODELS TRAINED ON $W_{\Theta,d}$

**Binary classification with logistic regression.** We consider the same setting as Bu et al. (2019, §VI): each data point $Z = (X, Y)$ consist of features $X \in \mathbb{R}^s$ and labels $Y \in \{0, 1\}$, $Y$ is uniformly distributed in $\{0, 1\}$, and $X|Y \sim \mathcal{N}(\mu_Y, 4\boldsymbol{I}_s)$ with $\mu_0 = (-1, \ldots, -1)$ and $\mu_1 = (1, \ldots, 1)$. We use a linear classifier and evaluate the generalization error based on the loss function $\ell(w, z) = \mathbf{1}_{\hat{y} \neq y}$, where $\hat{y}$ is the prediction of input $x$ defined as $\hat{y} \triangleq \mathbf{1}_{\bar{w}^T x + w_0 \geq 0}$, $\forall w = (\bar{w}, w_0) \in \mathbb{R}^{s+1}$. We train a logistic regression on $W_{\Theta,d}$ and estimate the generalization error. Since $\ell$ is bounded by $C = 1$, we approximate the generalization error bound given by Theorem 3.2 for $d < D$, and (Bu et al., 2019, Proposition 1) for $d = D$. Figure 2 reports the results for $s = 20$ and different values of $n$ and $d$: we observe that our bound holds and accurately reflects the behavior of the generalization error against $(n, d)$. Besides, our methodology provides tighter bounds than (Bu et al., 2019), and the difference increases with decreasing $d$. On the other hand, the lower $d$, the lower generalization error and its bound, but the higher the test risk (Figure 2). This is consistent with prior empirical studies (Li et al., 2018) and explained by the fact that lower values of $d$ induce a more restrictive hypothesis space, thus make the model less expressive.

**Multiclass classification with NNs.** Next, we evaluate our generalization error bounds for neural networks trained on image classification. Denote by $f(w, x) \in \mathbb{R}^K$ the output of the NN parameterized by $w$ given an input image $x$, with $K > 1$ the number of classes. The loss is $\ell(w, z) = \mathbf{1}_{\hat{y} \neq y}$, with $\hat{y} = \arg\max_{i \in \{1, \ldots, K\}} [f(w, x)]_i$. We train fully-connected NNs to classify MNIST and CIFAR-10 datasets, with $D = 199\,210$ and $D = 656\,810$ respectively: implementation details are given in Appendix B.2. Given the high-dimensionality of this problem, obtaining an accurate estimation of $\mathsf{I}^\Theta(\Theta; S_n)$ can be costly. Therefore, we discretize $W'$ with the quantizer from Lotfi et al. (2022) and evaluate Theorem 3.1 with $\mathsf{I}^\Theta(W'; S_n)$ replaced by $\lceil d \times H(p) \rceil + L \times (16 + \lceil \log_2 d \rceil) + 2$, as discussed at the beginning of Section 4. Results are shown in Figure 3 and demonstrate that our methodology allow us to compute generalization bounds for NNs, while also maintaining performance for reasonable values of $d$, which is consistent with Li et al. (2018). Additional empirical results on MNIST and Iris are given in Appendix B.2.

### 4.2 ILLUSTRATION OF RATE-DISTORTION BOUNDS

We solve a binary classification task with the neural network $f(w, x) = (f_2 \circ \varphi \circ f_1)(w, x)$, where $x$ is the input data, $f_i(w, x) = w_i x + b_i$ for $i \in \{1, 2\}$, and $\varphi(t) = \mathbf{1}_{t>0}$ is the ReLU activation function. The loss is the binary cross-entropy, *i.e.*, for $w \in \mathbb{R}^D$, $z = (X, y) \in \mathbb{R}^s \times \{0, 1\}$, $\ell(w, z) = -y \log(\sigma(f(w, X))) + (1 - y) \log(1 - \sigma(f(w, X)))$, where $\sigma(t) \triangleq 1/(1 + e^{-t})$ is the sigmoid function. The conditions of Theorems 3.3 and 3.4 are satisfied in this setting: $\ell$ is bounded

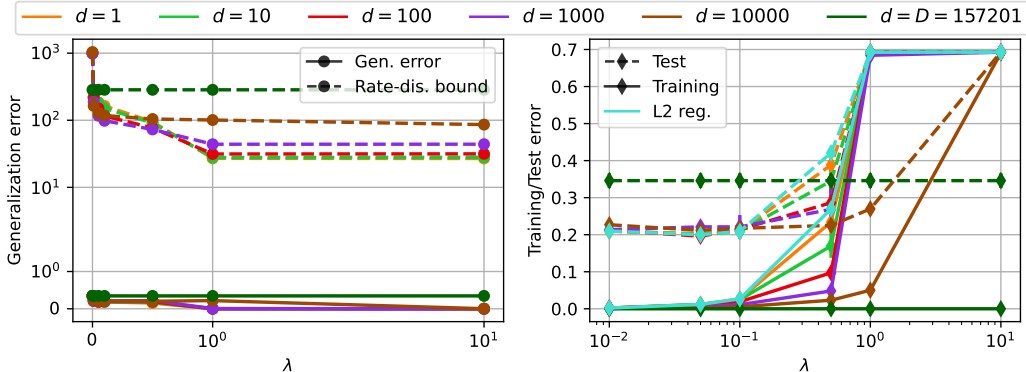

Figure 4: Influence of $(\lambda, d)$ on generalization errors and rate-distortion bounds *(left)*, and training and test errors *(right)* for a Lipschitz-constrained neural network on MNIST classification. Results are averaged over 5 runs.

since $f(w, X)$ admits a lower bound, and for any $z \in \mathbb{R}^{s+1}$, $\ell(\cdot, z)$ is Lipschitz-continuous. The explicit formulas of the bound $C$ and Lipschitz constant $L$ of the loss are given in Appendix B.3. Note that one can adjust the bound on $\ell$ (hence, the rate-distortion bound) by replacing $f(w, X)$ in the expression of $\ell(w, z)$ by $f(w, X) + \varepsilon$, where $\varepsilon \in \mathbb{R}$ is a hyperparameter.

Here, each sample $z$ corresponds to a pair of image $X \in \mathbb{R}^s$ and label $y \in \{0, 1\}$, where $y = 1$ if $X$ corresponds to a certain class (e.g., digit 1 for MNIST), $y = 0$ otherwise. For different $d$ and $\lambda$, we train on MNIST with only $n = 10$ samples, so that it is harder for the model to generalize well. We approximate the generalization error and our rate-distortion bound given in Theorem 3.4 with $\delta = 1/n$: see Figure 4. For any $d < D$, both the generalization error and rate-distortion bound decrease with increasing $\lambda$, as expected: higher values of $\lambda$ yield solutions $W$ with smaller $\|W - \Theta\Theta^\top W\|$, hence the model is more compressible (thus generalizes better) and the bound (9) is lower. We analyze the impact of $\lambda$ and $d$ on the test risk: Figure 4 shows that when $\lambda$ exceeds a certain threshold (which depends on $d$), the test risk increases, thus illustrating the trade-off between low generalization error and test risk for compressible models. We also observe that for some $(\lambda, d)$, the test risk is lower than the one returned by no regularization ($\lambda = 0$ or $d = D$) or the traditional L2 regularization ($\lambda\|W\|$ is added to the objective), which can both be seen as particular cases of our regularization technique. This suggests that for carefully chosen $\lambda$ and $d$, our methodology can be beneficial in tightening the information-theoretic generalization bounds, while improving the model's performance.

## 5 CONCLUSION

In this work, we combined recent empirical schemes for finding compressed models, including NNs, via random slicing with generalization bounds based on input-output MI. Our results indicate that architectures that are amenable to this compression scheme yield tighter information-theoretic generalization bounds. We also explore a notion of *approximate compressibility*, *i.e.*, *rate-distortion*, where the learned parameters are close to a quantization of the compressed subspace but do not lie on it exactly. This framework provides more flexibility in the trained model, allowing it to maintain good training error for even smaller (approximate) projection dimension $d$, ensuring that the resulting generalization bounds are as tight as possible, and allowing for clear analytical bounds on the MI to be used in place of difficult-to-compute MI estimates. This rate-distortion framework also motivated a weight regularization approach that encourages trained NNs to be as approximately compressible as possible to ensure that our bound is small in practice, while also providing empirical benefits in observed test performance itself. Future work includes a more detailed exploration of strategies for using our bounds to help inform selection and design of NN architectures in practice, and exploring bounds and regularizers based on other successful compression strategies for NNs, as discussed in the introduction.

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

## A  POSTPONED PROOFS FOR SECTION 3

### A.1  PROOF OF THEOREM A.2

Consider three random variables $X \in \mathrm{X}$, $Y \in \mathrm{Y}$ and $U \in \mathrm{U}$. Denote by $P_{X,Y,U}$ their joint distribution and by $P_X, P_Y, P_U$ the marginals. Let $\tilde{X}$ (respectively, $\tilde{Y}$) be an independent copy of $X$ (resp., $Y$) with joint distribution $P_{\tilde{X},\tilde{Y}} = P_{\tilde{X}} \otimes P_{\tilde{Y}}$. Given $U$, let $f^U : \mathrm{X} \times \mathrm{Y} \to \mathbb{R}$ be a mapping parameterized by $U$, and denote by $K_{f^U(\tilde{X},\tilde{Y})}$ the cumulant generating function of $f^U(\tilde{X},\tilde{Y})$, *i.e.* for $t \in \mathbb{R}$,

$$K_{f^U(\tilde{X},\tilde{Y})}(t) = \log \mathbb{E}\big[e^{t(f^U(\tilde{X},\tilde{Y}) - \mathbb{E}[f^U(\tilde{X},\tilde{Y})])}\big] \tag{10}$$

where the expectations are taken w.r.t. $P_{X|U} \otimes P_{Y|U}$.

**Lemma A.1.** *Suppose that for any $U \sim P_U$, there exists $b_+ \in \mathbb{R}_+^* \cup \{+\infty\}$ and a convex function $\varphi_+(\cdot, U) : [0, b_+) \to \mathbb{R}$ such that $\varphi_+(0, U) = \varphi_+'(0, U) = 0$ and for $t \in [0, b_+)$, $K_{f^U(\tilde{X},\tilde{Y})}(t) \le \psi_+(t, U)$. Then,*

$$\mathbb{E}_{P_{X,Y,U}}[f^U(X, Y)] - \mathbb{E}_{P_{\tilde{X},\tilde{Y},U}}[f^U(\tilde{X}, \tilde{Y})] \le \mathbb{E}_{P_U}\left[\inf_{t \in [0, b_+)} \frac{\mathsf{I}^U(X; Y) + \psi_+(t, U)}{t}\right]. \tag{11}$$

*Suppose that for any $U \sim P_U$, there exists $b_- \in \mathbb{R}_+^* \cup \{+\infty\}$ and a convex function $\varphi_-(\cdot, U) : [0, b_-) \to \mathbb{R}$ such that $\varphi_-(0, U) = \varphi_-'(0, U) = 0$ and for $t \in (b_-, 0]$, $K_{f^U(\tilde{X},\tilde{Y})}(t) \le \psi_-(-t, U)$. Then,*

$$\mathbb{E}_{P_{\tilde{X},\tilde{Y},U}}[f^U(\tilde{X}, \tilde{Y})] - \mathbb{E}_{P_{X,Y,U}}[f^U(X, Y)] \le \mathbb{E}_{P_U}\left[\inf_{t \in [0, -b_-)} \frac{\mathsf{I}^U(X; Y) + \psi_-(t, U)}{t}\right]. \tag{12}$$

*Proof.* Let $U \sim P_U$. By Donsker-Varadhan variational representation,

$$\mathsf{I}^U(X; Y) = \mathbf{KL}(P_{(X,Y)|U} \| P_{X|U} \otimes P_{Y|U}) \tag{13}$$

$$= \sup_{g \in \mathcal{G}^U} \mathbb{E}_{P_{(X,Y)|U}}[g^U(X, Y)] - \log \mathbb{E}_{P_{X|U} \otimes P_{Y|U}}[e^{g^U(\tilde{X},\tilde{Y})}] \tag{14}$$

where $\mathcal{G}^U \triangleq \{g^U : \mathrm{X} \times \mathrm{Y} \to \mathbb{R} \text{ s.t. } \mathbb{E}_{P_{X|U} \otimes P_{Y|U}}[e^{g^U(\tilde{X},\tilde{Y})}] < \infty\}$. Therefore, for any $t \in [0, b_+)$,

$$\mathbf{KL}(P_{(X,Y)|U} \| P_{X|U} \otimes P_{Y|U}) \ge t \mathbb{E}[f^U(X, Y)] - \log \mathbb{E}[e^{t f^U(\tilde{X},\tilde{Y})}] \tag{15}$$

$$\ge t \left(\mathbb{E}[f^U(X, Y)] - \mathbb{E}[f^U(\tilde{X}, \tilde{Y})]\right) - \psi_+(t, U) \tag{16}$$

where (16) follows from assuming that for $t \in [0, b_+)$, $K_{f^U(\tilde{X},\tilde{Y})}(t) \le \psi_+(t, U)$. Hence,

$$\mathbb{E}[f^U(X, Y)] - \mathbb{E}[f^U(\tilde{X}, \tilde{Y})] \le \inf_{t \in [0, b_+)} \frac{\mathsf{I}^U(X; Y) + \psi_+(t, U)}{t}. \tag{17}$$

We obtain the final result (11) by taking the expectation of (17) over $P_U$.

We can prove analogously that (12) holds, assuming for $t \in [0, b_-)$, $K_{f^U(\tilde{X},\tilde{Y})}(t) \le \psi_-(-t, U)$.

$\square$

**Theorem A.2.** *Assume that for $\Theta \sim P_\Theta$, there exists $C_- \in \mathbb{R}_+^* \cup \{+\infty\}$ s.t. for $t \in (C_-, 0]$, $K_{\ell^\Theta(\tilde{W}',\tilde{Z})}(t) \le \psi_-(-t, \Theta)$, where $\psi_-(\cdot, \Theta)$ is convex and $\psi_-(0, \Theta) = \psi_-'(0, \Theta) = 0$. Then,*

$$\mathrm{gen}(\mu, \mathcal{A}^{(d)}) \le \frac{1}{n} \sum_{i=1}^n \mathbb{E}_{P_\Theta}\left[\inf_{t \in [0, -C_-)} \frac{\mathsf{I}^\Theta(W'; Z_i) + \psi_-(t, \Theta)}{t}\right]. \tag{18}$$

*Assume that for $\Theta \sim P_\Theta$, there exists $C_+ \in \mathbb{R}_+^* \cup \{+\infty\}$ s.t. for $t \in [0, C_+)$, $K_{\ell^\Theta(\tilde{W}',\tilde{Z})}(t) \le \psi_+(t, \Theta)$, where $\psi_+(\cdot, \Theta)$ is convex and $\psi_+(0, \Theta) = \psi_+'(0, \Theta) = 0$. Then,*

$$\mathrm{gen}(\mu, \mathcal{A}^{(d)}) \ge \frac{1}{n} \sum_{i=1}^n \mathbb{E}_{P_\Theta}\left[\inf_{t \in [0, C_+)} \frac{\mathsf{I}^\Theta(W'; Z_i) + \psi_+(t, \Theta)}{t}\right]. \tag{19}$$

*Proof of Theorem A.2.* The generalization error of $\mathcal{A}^{(d)}$ can be written as

$$\text{gen}(\mu, \mathcal{A}^{(d)}) = \frac{1}{n} \sum_{i=1}^{n} \left\{ \mathbb{E}_{P_{W'|\Theta} \otimes P_{\Theta} \otimes \mu}[\ell^{\Theta}(\tilde{W}', \tilde{Z}_i)] - \mathbb{E}_{P_{W'|\Theta, Z_i} \otimes P_{\Theta} \otimes \mu}[\ell^{\Theta}(W', Z_i)] \right\}. \quad (20)$$

Our final bounds (19) and (18) result from applying Lemma A.1 on each term of the sum in (20), *i.e.* with $X = W'$, $Y = Z_i$ and $f^U(X, Y) = \ell^{\Theta}(W', Z_i)$.

$\square$

## A.2 APPLICATIONS OF THEOREM A.2

We specify Theorem A.2 under different sub-Gaussian conditions on the loss. A random variable $X$ is said to be $\sigma$-sub-Gaussian (with $\sigma > 0$) if for any $t \in \mathbb{R}$,

$$\mathbb{E}[e^{t(X - \mathbb{E}[X])}] \leq e^{\sigma^2 t^2 / 2}. \quad (21)$$

*Proof of Theorem 3.1.* Define $h^{\Theta}(w', s) = (1/n) \sum_{i=1}^{n} \ell^{\Theta}(w', z_i)$ for $w' \in \mathbb{R}^d$, $s = (z_1, \ldots, z_n) \in \mathsf{Z}^n$ and $\Theta \in \mathbb{R}^{D \times d}$ s.t. $\Theta^{\top} \Theta = \boldsymbol{I}_d$. The generalization error of $\mathcal{A}^{(d)}$ can be written as,

$$\text{gen}(\mu, \mathcal{A}^{(d)}) = \mathbb{E}_{P_{W'|\Theta} \otimes P_{\Theta} \otimes \mu^{\otimes n}} \left[ h^{\Theta}(\tilde{W}', \tilde{S}_n) \right] - \mathbb{E}_{P_{W'|Z_i, \Theta} \otimes P_{\Theta} \otimes \mu^{\otimes n}} \left[ h^{\Theta}(W', S_n) \right]. \quad (22)$$

Since we assume that $\ell^{\Theta}(w', Z)$ is $\sigma$-sub-Gaussian under $Z \sim \mu$ for all $w'$ and $\Theta$, and $Z_1, \ldots, Z_n$ are i.i.d, then $h^{\Theta}(w', S_n)$ is $\sigma/\sqrt{n}$-sub-Gaussian under $S_n \sim \mu^{\otimes n}$ for all $w'$ and $\Theta$. Therefore, $h^{\Theta}(\tilde{W}', \tilde{S}_n)$ is $\sigma/\sqrt{n}$-sub-Gaussian under $(\tilde{W}', S_n) \sim P_{W'|\Theta} \otimes \mu^{\otimes n}$ for all $\Theta$, and for $t \in \mathbb{R}$,

$$K_{h^{\Theta}(\tilde{W}', \tilde{S}_n)}(t) \leq \frac{\sigma^2 t^2}{2n}. \quad (23)$$

We conclude by applying Lemma A.1 with $X = W'$, $Y = S_n$, $U = \Theta$ and $f^U(X, Y) = h^{\Theta}(W', S_n)$, and the fact that,

$$\inf_{t > 0} \frac{\mathsf{I}^{\Theta}(W'; S_n) + \sigma^2 t^2 / (2n)}{t} = \sqrt{\frac{2\sigma^2}{n} \mathsf{I}^{\Theta}(W'; S_n)}. \quad (24)$$

$\square$

*Proof of Theorem 3.2.* Let $\Theta \in \mathbb{R}^{D \times d}$ s.t. $\Theta^{\top} \Theta = \boldsymbol{I}_d$. Since $\ell^{\Theta}(\tilde{W}', \tilde{Z})$ is $\sigma_{\Theta}$-sub-Gaussian under $(\tilde{W}', \tilde{Z}) \sim P_{W'} \otimes \mu$, then for any $t \in \mathbb{R}$, $K_{\ell^{\Theta}(\tilde{W}', \tilde{Z})}(t) \leq \sigma_{\Theta}^2 t^2 / 2$. We conclude by applying Theorem A.2 and the fact that for $i \in \{1, \ldots, n\}$,

$$\inf_{t > 0} \frac{\mathsf{I}^{\Theta}(W'; Z_i) + \sigma_{\Theta}^2 t^2 / 2}{t} = \sqrt{2\sigma_{\Theta}^2 \mathsf{I}^{\Theta}(W'; Z_i)}. \quad (25)$$

$\square$

**Corollary A.3.** *Assume that for any $\Theta \sim P_{\Theta}$, $\ell^{\Theta}(\tilde{W}', \tilde{Z}) \leq C$ almost surely. Then,*

$$|\text{gen}(\mu, \mathcal{A}^{(d)})| \leq \frac{C}{n} \sum_{i=1}^{n} \mathbb{E}_{P_{\Theta}} \left[ \sqrt{\frac{\mathsf{I}^{\Theta}(W'; Z_i)}{2}} \right]. \quad (26)$$

*Proof of Corollary A.3.* Since for any $\Theta \sim P_{\Theta}$, $\ell^{\Theta}(\tilde{W}', \tilde{Z}) \leq C$ almost surely, then by Hoeffding's lemma, we have for all $t \in \mathbb{R}$,

$$\mathbb{E}_{P_{W'|\Theta} \otimes \mu} \left[ e^{t\{\ell^{\Theta}(\tilde{W}', \tilde{Z}) - \mathbb{E}_{P_{W'|\Theta} \otimes \mu}[\ell^{\Theta}(\tilde{W}', \tilde{Z})]\}} \right] \leq e^{C^2 t^2 / 8}. \quad (27)$$

Therefore, $K_{\ell^{\Theta}(\tilde{W}', \tilde{Z})}(t) \leq C^2 t^2 / 8$. We conclude by applying Lemma A.1 and the fact that for $i \in \{1, \ldots, n\}$,

$$\inf_{t > 0} \frac{\mathsf{I}^{\Theta}(W'; Z_i) + C^2 t^2 / 8}{t} = C \sqrt{\frac{\mathsf{I}^{\Theta}(W'; Z_i)}{2}}. \quad (28)$$

$\square$

### A.3 DETAILED DERIVATIONS FOR GAUSSIAN MEAN ESTIMATION

**Problem statement.** The loss function is defined for any $(w, z) \in \mathbb{R}^D \times \mathbb{R}^D$ as $\ell(w, x) = \|w - z\|^2$. Let $Z_1, \ldots, Z_n$ be $n$ random variables i.i.d. from $\mathcal{N}(\mathbf{0}_D, \mathbf{I}_D)$. Let $d \leq D$ and $\Theta \sim P_\Theta$ s.t. $\Theta^\top \Theta = \mathbf{I}_d$. Consider a model $\mathcal{A}^{(d)}$ whose objective is $\arg\min_{w \in W_{\Theta,d}} \widehat{\mathcal{R}}_n(w)$ where the empirical risk is defined for $w \in \mathbb{R}^D$ as $\widehat{\mathcal{R}}_n(w) = \frac{1}{n} \sum_{i=1}^n \|w - Z_i\|^2$. This is equivalent to solving $\arg\min_{w' \in \mathbb{R}^d} \widehat{\mathcal{R}}_n^\Theta(w')$, where

$$\forall w' \in \mathbb{R}^d, \ \ \widehat{\mathcal{R}}_n^\Theta(w') = \frac{1}{n} \sum_{i=1}^n \|\Theta w' - Z_i\|^2. \tag{29}$$

The gradient of (59) with respect to $w'$ is,

$$\nabla_{w'} \widehat{\mathcal{R}}_n^\Theta(w) = \frac{2}{n} \sum_{i=1}^n \Theta^\top (\Theta w' - Z_i), \tag{30}$$

and solving $\nabla_{w'} \widehat{\mathcal{R}}_n^\Theta(w) = 0$ yields $(\Theta^\top \Theta)w' = \Theta^\top \bar{Z}$ where $\bar{Z} \triangleq (1/n) \sum_{i=1}^n Z_i$. Since $\Theta^\top \Theta = \mathbf{I}_d$, we conclude that the minimizer of (59) is $W' = \Theta^\top \bar{Z}$.

**Generalization error.** We recall that the generalization error of $\mathcal{A}^{(d)}$ is defined as,

$$\mathrm{gen}(\mu, \mathcal{A}^{(d)}) = \mathbb{E}[\mathcal{R}^\Theta(W') - \widehat{\mathcal{R}}_n^\Theta(W')] \tag{31}$$

where the expectation is computed with respect to $P_{W'|\Theta, S_n} \otimes P_\Theta \otimes \mu^{\otimes n}$. Since $W' = \Theta^\top \bar{Z}$, $\mathrm{gen}(\mu, \mathcal{A}^{(d)})$ can be written as

$$\mathrm{gen}(\mu, \mathcal{A}^{(d)}) = \mathbb{E}_{(S_n, \Theta) \sim \mu^{\otimes n} \otimes P_\Theta} \left[ \mathbb{E}_{\tilde{Z} \sim \mu}[\|\Theta \Theta^\top \bar{Z} - \tilde{Z}\|^2] - \frac{1}{n} \sum_{i=1}^n \|\Theta \Theta^\top \bar{Z} - Z_i\|^2 \right] \tag{32}$$

Since $Z_1, \ldots, Z_n$ are $n$ i.i.d. samples from $\mathcal{N}(\mathbf{0}_D, \mathbf{I}_D)$ and $\Theta^\top \Theta = \mathbf{I}_d$, then $P_{\Theta^\top \bar{Z}|\Theta} = \mathcal{N}(\mathbf{0}_d, (1/n)\mathbf{I}_d)$ and we have

$$\mathbb{E}_{\mu^{\otimes n} \otimes P_\Theta}[\|\Theta \Theta^\top \bar{Z}\|^2] = \mathbb{E}_{\mu^{\otimes n} \otimes P_\Theta}[\mathrm{Tr}((\Theta \Theta^\top \bar{Z})^\top (\Theta \Theta^\top \bar{Z}))] \tag{33}$$

$$= \mathbb{E}_{\mu^{\otimes n} \otimes P_\Theta}[\mathrm{Tr}(\bar{Z}^\top \Theta \Theta^\top \Theta \Theta^\top \bar{Z})] \tag{34}$$

$$= \mathrm{Tr}(\mathbb{E}_{\mu^{\otimes n} \otimes P_\Theta}[\Theta^\top \bar{Z}(\Theta^\top \bar{Z})^\top]) \tag{35}$$

$$= \frac{d}{n}. \tag{36}$$

Besides, for $i \in \{1, \ldots, n\}$, $\mathbb{E}[\|Z_i\|^2] = \mathrm{Tr}(\mathbb{E}[Z_i Z_i^\top]) = D$, and

$$\mathbb{E}[(\Theta \Theta^\top \bar{Z})^\top Z_i] = \frac{1}{n} \sum_{j=1}^n \mathbb{E}[Z_j^\top \Theta \Theta^\top Z_i] \tag{37}$$

$$= \frac{1}{n} \sum_{j=1}^n \mathrm{Tr}(\mathbb{E}[\Theta^\top Z_i (\Theta^\top Z_j)^\top]) \tag{38}$$

$$= \frac{1}{n} \mathrm{Tr}(\mathbb{E}[\Theta^\top Z_i (\Theta^\top Z_i)^\top]) \tag{39}$$

$$= \frac{d}{n}. \tag{40}$$

Equations (39) to (40) can be justified as follows. Since $Z_i \sim \mathcal{N}(\mathbf{0}_D, \mathbf{I}_D)$, the conditional distribution of $\Theta^\top Z_i$ given $\Theta$ is $\mathcal{N}(\mathbf{0}_d, \Theta^\top \Theta)$, and $\Theta^\top \Theta = \mathbf{I}_d$ by definition. Therefore, $\mathbb{E}[\Theta^\top Z_i (\Theta^\top Z_i)^\top] = \mathbb{E}[\mathbb{E}[\Theta^\top Z_i (\Theta^\top Z_i)^\top | \Theta]] = \mathbf{I}_d$. We conclude that $\mathrm{Tr}(\mathbb{E}[\Theta^\top Z_i (\Theta^\top Z_i)^\top]) = d$.

We thus obtain,

$$\mathbb{E}[\widehat{\mathcal{R}}_n^{\Theta}(W')] = \mathbb{E}_{(S_n,\Theta)\sim\mu^{\otimes n}\otimes P_\Theta}\left[\frac{1}{n}\sum_{i=1}^n \|\Theta\Theta^\top\bar{Z} - Z_i\|^2\right] \tag{41}$$

$$= \mathbb{E}_{(S_n,\Theta)\sim\mu^{\otimes n}\otimes P_\Theta}\left[\frac{1}{n}\sum_{i=1}^n \|\Theta\Theta^\top\bar{Z}\|^2 - 2(\Theta\Theta^\top\bar{Z})^\top Z_i + \|Z_i\|^2\right] \tag{42}$$

$$= D - \frac{d}{n}\,. \tag{43}$$

Indeed, by the linearity of expectation, (42) simplifies as

$$\mathbb{E}[\widehat{\mathcal{R}}_n^{\Theta}(W')] = \mathbb{E}_{\mu^{\otimes n}\otimes P_\Theta}[\|\Theta\Theta^\top\bar{Z}\|^2] - \frac{2}{n}\sum_{i=1}^n \mathbb{E}_{\mu^{\otimes n}\otimes P_\Theta}[(\Theta\Theta^\top\bar{Z})^\top Z_i] + \frac{1}{n}\sum_{i=1}^n \mathbb{E}_{\mu}[\|Z_i\|^2] \tag{44}$$

Since $(Z_i)_{i=1}^n$ are i.i.d. from $\mathcal{N}(\mathbf{0}_D, \mathbf{I}_D)$, we proved that $\mathbb{E}_{\mu^{\otimes n}\otimes P_\Theta}[\|\Theta\Theta^\top\bar{Z}\|^2] = \frac{d}{n}$ (eq. (36)) and $\mathbb{E}[(\Theta\Theta^\top\bar{Z})^\top Z_i] = \frac{d}{n}$ (eq. (40)). Additionally,

$$\mathbb{E}_{\mu}[\|Z_i\|^2] = \mathbb{E}_{\mu}[\mathrm{Tr}(\|Z_i\|^2)] = \mathbb{E}_{\mu}[\mathrm{Tr}(Z_i Z_i^\top)] = \mathrm{Tr}(\mathbb{E}_{\mu}[Z_i Z_i^\top]) = \mathrm{Tr}(\mathbf{I}_D) = D \tag{45}$$

Plugging these identities in (44) yields (43).

On the other hand,

$$\mathbb{E}_{(S_n,\Theta,\tilde{Z})\sim\mu^{\otimes n}\otimes P_\Theta\otimes\mu}[(\Theta\Theta^\top\bar{Z})^\top\tilde{Z}] = \mathbb{E}[\Theta\Theta^\top\bar{Z}]^\top\mathbb{E}[\tilde{Z}] = 0\,, \tag{46}$$

therefore,

$$\mathbb{E}[\mathcal{R}^{\Theta}(W')] = \mathbb{E}_{(S_n,\Theta)\sim\mu^{\otimes n}\otimes P_\Theta}\mathbb{E}_{\tilde{Z}\sim\mu}[\|\Theta\Theta^\top\bar{Z} - \tilde{Z}\|^2] \tag{47}$$

$$= \mathbb{E}_{(S_n,\Theta)\sim\mu^{\otimes n}\otimes P_\Theta}\mathbb{E}_{\tilde{Z}\sim\mu}[\|\Theta\Theta^\top\bar{Z}\|^2 - 2(\Theta\Theta^\top\bar{Z})^\top\tilde{Z} + \|\tilde{Z}\|^2] \tag{48}$$

$$= D + \frac{d}{n}\,. \tag{49}$$

By plugging (43) and (49) in (32), we conclude that $\mathrm{gen}(\mu, \mathcal{A}^{(d)}) = 2d/n$.

**Generalization error bound.** We apply Theorem A.2 to bound the generalization error. To this end, we need to bound the cumulant generating function of $\ell^{\Theta}(\tilde{W}', \tilde{Z}) = \|\Theta\Theta^\top\bar{Z} - \tilde{Z}\|^2$ given $\Theta$.

Since $(Z_1, \ldots, Z_n, \tilde{Z}) \sim \mu^{\otimes n} \otimes \mu$ with $\mu = \mathcal{N}(\mathbf{0}_D, \mathbf{I}_D)$, then, given $\Theta$, one has $\Theta^\top\bar{Z} \sim \mathcal{N}(\mathbf{0}_d, (1/n)\mathbf{I}_d)$ and $(\Theta\Theta^\top\bar{Z} - \tilde{Z}) \sim \mathcal{N}(\mathbf{0}_D, \Sigma_\Theta)$ with $\Sigma_\Theta = \Theta\Theta^\top/n + \mathbf{I}_D$. Therefore, for $d < D$, $\ell^{\Theta}(\tilde{W}', \tilde{Z}) = \|\Theta\Theta^\top\bar{Z} - \tilde{Z}\|^2$ is the sum of squares of $D$ dependent Gaussian random variables, which can equivalently be written as

$$\ell^{\Theta}(\tilde{W}', \tilde{Z}) = \sum_{k=1}^D \lambda_{\Theta,k} U_{\Theta,k}^2\,, \tag{50}$$

$$U_\Theta = P\Sigma_\Theta^{-1/2}(\Theta W' - \tilde{Z}) \tag{51}$$

where $P \in \mathbb{R}^{D\times D}$ and $\lambda_\Theta = (\lambda_{\Theta,1}, \ldots, \lambda_{\Theta,D}) \in \mathbb{R}^D$ come from the eigendecomposition of $\Sigma_\Theta$, i.e. $\Sigma_\Theta = P\Lambda P^\top$ with $\Lambda = \mathrm{diag}(\lambda_\Theta)$. As a consequence, $U_\Theta \sim \mathcal{N}(\mathbf{0}_D, \mathbf{I}_D)$. Note that, since $\Sigma_\Theta$ is positive definite, $P$ is orthogonal and for any $k \in \{1, \ldots, D\}$, $\lambda_{\Theta,k} > 0$.

By (50), $\ell^{\Theta}(\tilde{W}', \tilde{Z})$ is a linear combination of independent chi-square variables, each with 1 degree of freedom. Therefore, $\ell^{\Theta}(\tilde{W}', \tilde{Z})$ is distributed from a generalized chi-square distribution, and its CGF is given by,

$$\forall t \le \frac{1}{2}\min_{k\in\{1,\ldots,D\}}\lambda_{\Theta,k}, \quad K_{\ell^{\Theta}(\tilde{W}',\tilde{Z})}(t) = -t\sum_{k=1}^D \lambda_{\Theta,k} - \frac{1}{2}\sum_{k=1}^D \log(1 - 2\lambda_{\Theta,k}t) \tag{52}$$

$$= \frac{1}{2}\sum_{k=1}^D [-2\lambda_{\Theta,k}t - \log(1 - 2\lambda_{\Theta,k}t)]\,. \tag{53}$$

Since for any $s < 0$, $-s - \log(1 - s) \le s^2/2$, we deduce that

$$\forall t < 0, \quad K_{\ell\Theta(\tilde{W}', \tilde{Z})}(t) \le \frac{1}{2} \sum_{k=1}^{D} \frac{(2\lambda_{\Theta,k} t)^2}{2} = \|\lambda_\Theta\|^2 t^2 \,. \tag{54}$$

Since $\mathrm{rank}(\Theta\Theta^\top) = \mathrm{rank}(\Theta^\top\Theta)$ and $\Theta^\top\Theta = \boldsymbol{I}_d$, then $\mathrm{rank}(\Theta\Theta^\top) = d$. Besides, $\Theta\Theta^\top$ and $\Theta^\top\Theta$ share the same non-zero eigenvalues. Therefore, $\Theta\Theta^\top$ has $d$ eigenvalues equal to 1, and $(D - d)$ eigenvalues equal to 0, thus $\Theta^\top\Theta/n + \boldsymbol{I}_d$ has $d$ eigenvalues equal to $1 + 1/n$ and and $(D - d)$ eigenvalues equal to 1, and

$$\|\lambda_\Theta\|^2 = d\left(1 + \frac{1}{n}\right)^2 + (D - d) \,. \tag{55}$$

By combining Theorem A.2 with (54) and (55), we obtain

$$\mathrm{gen}(\mu, \mathcal{A}^{(d)}) \le \frac{2}{n} \sqrt{d\left(1 + \frac{1}{n}\right)^2 + (D - d)} \sum_{i=1}^{n} \mathbb{E}_{P_\Theta}\left[\sqrt{\mathsf{I}^\Theta(W'; Z_i)}\right] \tag{56}$$

Applying Jensen's inequality on (56) and the fact that $W' = \Theta^\top W$ with $W = \arg\min_{w \in \mathbb{R}^D} \widehat{\mathcal{R}}_n(w) = \bar{Z}$ finally yields,

$$\mathrm{gen}(\mu, \mathcal{A}^{(d)}) \le \frac{2}{n} \sqrt{d\left(1 + \frac{1}{n}\right)^2 + (D - d)} \sum_{i=1}^{n} \sqrt{\mathsf{SI}_d^{(1)}(W; Z_i)} \,. \tag{57}$$

## A.4 DETAILED DERIVATIONS FOR LINEAR REGRESSION

**Problem statement.** Consider $n$ i.i.d. samples $(x_1, \ldots, x_n)$ and a response variable $y = (y_1, \ldots, y_n)$, where $x_i \in \mathbb{R}^D$ and $y_i \in \mathbb{R}$. Consder a learning algorithm $\mathcal{A}^{(d)}$ whose objective is $\arg\min_{w \in \mathsf{W}_{\Theta,d}} \widehat{\mathcal{R}}_n(w)$, with

$$\forall w \in \mathbb{R}^D, \quad \widehat{\mathcal{R}}_n(w) = \frac{1}{n} \sum_{i=1}^{n} (y_i - x_i^\top w)^2 = \frac{1}{n} \|y - Xw\|^2 \,. \tag{58}$$

where $X \in \mathbb{R}^{n \times D}$ is the design matrix. This objective is equivalent to finding $W' = \arg\min_{w' \in \mathbb{R}^d} \widehat{\mathcal{R}}_n^\Theta(w')$, where

$$\forall w' \in \mathbb{R}^d, \quad \widehat{\mathcal{R}}_n^\Theta(w') = \frac{1}{n} \|y - X\Theta w'\|^2 \,. \tag{59}$$

We assume the problem is over-determined, *i.e.* $D \le n$. Solving $\nabla_{w'} \widehat{\mathcal{R}}_n^\Theta(w') = 0$ yields

$$W' = (\Theta X^\top X \Theta^\top)^{-1} \Theta X^\top y \,. \tag{60}$$

On the other hand, we know that the solution of $\arg\min_{w \in \mathbb{R}^D} \widehat{\mathcal{R}}_n(w)$ is the ordinary least squares (OLS) estimator, given by

$$W = (X^\top X)^{-1} X^\top y \,. \tag{61}$$

Hence, by (61) with (60), we deduce that

$$W' = (\Theta X^\top X \Theta^\top)^{-1} \Theta (X^\top X) W \tag{62}$$

**Generalization error.** In the remainder of this section, we assume that $X$ is deterministic and there exists $W^\star \in \mathbb{R}^D$ such that $y_i = x_i^\top W^\star + \varepsilon_i$ where $(\varepsilon_i)_{i=1}^n$ are i.i.d. from $\mathcal{N}(0, \sigma^2)$. By using similar techniques as in Appendix A.3, one can show that $\mathrm{gen}(\mu, \mathcal{A}^{(d)}) = 2\sigma^2 d/n$.

**Generalization error bound.** Since $y_i \sim \mathcal{N}(x_i^\top W^\star, \sigma^2)$, and by (60),

$$x_i^\top \Theta^\top W' \sim \mathcal{N}(x_i^\top \Theta_X W^\star, \sigma^2 x_i^\top \Theta^\top [\Theta X^\top X \Theta^\top]^{-1} \Theta x_i) \tag{63}$$

where $\Theta_X = \Theta^\top (\Theta X^\top X \Theta^\top)^{-1} \Theta (X^\top X) \in \mathbb{R}^{D \times D}$. Therefore,

$$(\tilde{y}_i - x_i^\top \Theta^\top \tilde{W}') \sim \mathcal{N}(x_i^\top (\mathbf{I}_D - \Theta_X) W^\star, \sigma^2 (1 + x_i^\top \Theta^\top [\Theta X^\top X \Theta^\top]^{-1} \Theta x_i)), \tag{64}$$

and

$$\ell^\Theta(\tilde{W}', \tilde{y}_i) \sim \sigma_i^2 \chi^2(1, \lambda_i), \tag{65}$$

where $\sigma_i^2 = \sigma^2(1 + x_i^\top \Theta^\top [\Theta X^\top X \Theta^\top]^{-1} \Theta x_i)$, $\lambda_i = (x_i^\top (\mathbf{I}_D - \Theta_X) W^\star)^2$ and $\chi^2(k, \lambda)$ denotes the noncentral chi-squared distribution with $k$ degrees of freedom and noncentrality parameter $\lambda$. Hence, the moment-generating function of $\ell^\Theta(\tilde{W}', \tilde{y}_i)$ is

$$\forall t < \frac{1}{2\sigma_i^2}, \quad \mathbb{E}\big[e^{t\,\ell^\Theta(\tilde{W}', \tilde{y}_i)}\big] = \frac{e^{(\lambda_i \sigma_i^2 t)/(1 - 2\sigma_i^2 t)}}{\sqrt{1 - 2\sigma_i^2 t}} \tag{66}$$

and its expectation is $\mathbb{E}[\ell^\Theta(\tilde{W}', \tilde{y}_i)] = \sigma_i^2(1 + \lambda_i)$. Therefore, for $t < 1/(2\sigma_i^2)$ and $u_i = 2\sigma_i^2 t$,

$$K_{\ell^\Theta(\tilde{W}', \tilde{y}_i)}(t) = \frac{\lambda_i u_i}{2(1 - u_i)} - \frac{1}{2}\log(1 - u_i) - \frac{1}{2}(1 + \lambda_i)u_i \tag{67}$$

$$= \frac{1}{2}\{-\log(1 - u_i) - u_i\} + \frac{\lambda_i u_i^2}{2(1 - u_i)}. \tag{68}$$

Since $-\log(1 - x) - x \le x^2/2$ for $x < 0$, we deduce that for $t < 0$,

$$K_{\ell^\Theta(\tilde{W}', \tilde{y}_i)}(t) \le \frac{u_i^2}{4} + \frac{\lambda_i u_i^2}{2(1 - u_i)} \tag{69}$$

$$= \sigma_i^4 t^2 + \frac{2\lambda_i \sigma_i^4 t^2}{1 - 2\sigma_i^2 t}. \tag{70}$$

By applying Theorem A.2, we conclude that

$$\mathrm{gen}(\mu, \mathcal{A}^{(d)}) \le \frac{1}{n}\sum_{i=1}^n \mathbb{E}_\Theta\left[\inf_{t>0} \frac{\mathsf{I}(W'; y_i) + \sigma_i^4 t^2\big(1 + 2\lambda_i(1 + 2\sigma_i^2 t)^{-1}\big)}{t}\right]. \tag{71}$$

By (62), $W'$ is the projection of $W$ along $\phi(\Theta, X) \triangleq (\Theta X^\top X \Theta^\top)^{-1} \Theta (X^\top X)$. The right-hand side term in (71) can thus be interpreted as a generalized SMI with a non-isotropic slicing distribution that depends on the fixed $X$.

As $d$ converges to $D$, $\lambda = (\lambda_1, \ldots, \lambda_n) \in \mathbb{R}^n$ converges to $\mathbf{0}_n$. Indeed, consider the compact singular value decomposition (SVD) of $X\Theta^\top$, *i.e.* $X\Theta^\top = USV^\top$ where $S \in \mathbb{R}^{d \times d}$ is diagonal, $U \in \mathbb{R}^{n \times d}$, $V \in \mathbb{R}^{d \times m}$ s.t. $U^\top U = V^\top V = \mathbf{I}_d$. By using the pseudo-inverse expression of SVD,

$$X\Theta_X = X\Theta^\top (\Theta X^\top X \Theta^\top)^{-1} \Theta (X^\top X) \tag{72}$$

$$= USV^\top V S^{-1} U^\top X \tag{73}$$

$$= UU^\top X \tag{74}$$

Therefore, $\sqrt{\lambda} = X(\mathbf{I}_D - UU^\top)W^\star$. Since $U^\top U = \mathbf{I}_d$ with $U \in \mathbb{R}^{n \times d}$, then $\mathbf{I}_D - UU^\top$ has $(D - d)$ eigenvalues equal to 1 and $d$ eigenvalues equal to 0. Hence, $\lambda$ converges to $\mathbf{0}_n$ as $d \to D$.

## A.5 PROOFS FOR SECTION 3.3

*Proof of Theorem 3.3.* By the triangle inequality, for any pair of models $(\mathcal{A}, \mathcal{A}')$,

$$|\mathrm{gen}(\mu, \mathcal{A})| \le |\mathrm{gen}(\mu, \mathcal{A}) - \mathrm{gen}(\mu, \mathcal{A}')| + |\mathrm{gen}(\mu, \mathcal{A}')|. \tag{75}$$

Consider $\mathcal{A} : \mathsf{Z}^n \to \mathsf{W}$ and $\mathcal{A}' : \mathsf{Z}^n \to \mathsf{W}_{\Theta,d}$ such that $\mathcal{A}(S_n) = W$ may depend on $\Theta \sim P_\Theta$, and $\mathcal{A}'(S_n) = \Theta(\Theta^\top W)$. On the one hand, by applying Lemma A.1 with $X = \Theta^\top W$, $Y = Z_i$, $U = \Theta$ and $f^U(X, Y) = \ell^\Theta(\Theta^\top W, Z_i)$, we obtain

$$|\mathrm{gen}(\mu, \mathcal{A}')| \le \frac{C}{n}\sum_{i=1}^n \mathbb{E}_{P_\Theta}\left[\sqrt{\frac{\mathsf{I}^\Theta(\Theta^\top W; Z_i)}{2}}\right]. \tag{76}$$

On the other hand, by the definition of the generalization error, one can show that

$$|\text{gen}(\mu, \mathcal{A}) - \text{gen}(\mu, \mathcal{A}')| = |\mathbb{E}[\mathcal{R}(W) - \widehat{\mathcal{R}}_n(W)] - \mathbb{E}[\mathcal{R}^\Theta(\Theta^\top W) - \widehat{\mathcal{R}}_n^\Theta(\Theta^\top W)]| \qquad (77)$$

$$\leq |\mathbb{E}[\mathcal{R}(W) - \mathcal{R}^\Theta(\Theta^\top W)]| + |\mathbb{E}[\widehat{\mathcal{R}}_n(W) - \widehat{\mathcal{R}}_n^\Theta(\Theta^\top W)]| \qquad (78)$$

where the expectations are computed over $P_{W|\Theta, S_n} \otimes P_\Theta \otimes \mu^{\otimes n}$. Besides,

$$|\mathbb{E}[\mathcal{R}(W) - \mathcal{R}^\Theta(\Theta^\top W)]| = |\mathbb{E}_{P_{W|\Theta} \otimes P_\Theta \otimes \mu}[\ell(\tilde{W}, \tilde{Z}) - \ell(\Theta\Theta^\top \tilde{W}, \tilde{Z})]| \qquad (79)$$

$$\leq \mathbb{E}_{P_{W|\Theta} \otimes P_\Theta \otimes \mu} |\ell(W, Z) - \ell(\Theta\Theta^\top W, Z)| \qquad (80)$$

$$\leq L \mathbb{E}_{P_{W|\Theta} \otimes P_\Theta} \|W - \Theta\Theta^\top W\|, \qquad (81)$$

where (79) follows from the definition of the population risks $\mathcal{R}(w)$ and $\mathcal{R}^\Theta(\Theta^\top w)$, and (81) results from the assumption that $\ell(\cdot, z) : \mathrm{W} \to \mathbb{R}_+$ is $L$-Lipschitz for all $z \in \mathrm{Z}$.

Using similar arguments, one can show that

$$|\mathbb{E}[\widehat{\mathcal{R}}_n(W) - \widehat{\mathcal{R}}_n^\Theta(\Theta^\top W)]| \leq L \mathbb{E}_{P_{W|\Theta} \otimes P_\Theta} \|W - \Theta\Theta^\top W\|, \qquad (82)$$

and we conclude that

$$|\text{gen}(\mu, \mathcal{A}) - \text{gen}(\mu, \mathcal{A}')| \leq 2L \mathbb{E}_{P_{W|\Theta} \otimes P_\Theta} \|W - \Theta\Theta^\top W\|. \qquad (83)$$

The final result follows from bounding (75) using (76) and (83).

$\square$

*Proof of Theorem 3.4.* Consider $\mathcal{A} : \mathrm{Z}^n \to \mathrm{W}$ and $\mathcal{A}' : \mathrm{Z}^n \to \mathrm{W}_{\Theta,d}$ such that $\mathcal{A}(S_n) = W$ may depend on $\Theta \sim P_\Theta$, and $\mathcal{A}'(S_n) = \Theta\mathcal{Q}(\Theta^\top W)$. Using the same techniques as in the proof of Theorem 3.3, we obtain

$$|\text{gen}(\mu, \mathcal{A})| \leq 2L \mathbb{E}_{P_{W|\Theta} \otimes P_\Theta} \|W - \Theta\mathcal{Q}(\Theta^\top W)\| + |\text{gen}(\mu, \mathcal{A}')| \qquad (84)$$

$$\leq 2L \mathbb{E}_{P_{W|\Theta} \otimes P_\Theta} \|W - \Theta\mathcal{Q}(\Theta^\top W)\| + C \, \mathbb{E}_{P_\Theta} \left[ \sqrt{\frac{I^\Theta(\mathcal{Q}(\Theta^\top W); S_n)}{2n}} \right] \qquad (85)$$

where eq. (85) follows from applying Theorem 3.1.

Then, by using the triangle inequality, the fact that $\|\Theta\| = \|\Theta^\top \Theta\| = 1$, and the properties of $\mathcal{Q}$,

$$\mathbb{E}_{P_{W|\Theta} \otimes P_\Theta} \|W - \Theta\mathcal{Q}(\Theta^\top W)\| \qquad (86)$$

$$\leq \mathbb{E}_{P_{W|\Theta} \otimes P_\Theta} \|W - \Theta\Theta^\top W\| + \mathbb{E}_{P_{W|\Theta} \otimes P_\Theta} \|\Theta\Theta^\top W - \Theta\mathcal{Q}(\Theta^\top W)\| \qquad (87)$$

$$\leq \mathbb{E}_{P_{W|\Theta} \otimes P_\Theta} \|W - \Theta\Theta^\top W\| + \mathbb{E}_{P_{W|\Theta} \otimes P_\Theta} [\|\Theta\| \|\Theta^\top W - \mathcal{Q}(\Theta^\top W)\|] \qquad (88)$$

$$\leq \mathbb{E}_{P_{W|\Theta} \otimes P_\Theta} \|W - \Theta\Theta^\top W\| + \delta. \qquad (89)$$

Finally, since $\mathcal{Q}(\Theta^\top W)$ is a discrete random variable and $\|\Theta^\top W\| \leq M$, we use the same arguments as in Section 3.1 to bound $I^\Theta(\mathcal{Q}(\Theta^\top W); S_n)$ by $d \log(2M\sqrt{d}/\delta)$.

$\square$

# B   Additional Experimental Details for Section 4

## B.1   Methodological Details

**Architecture for MINE.**   In all our experiments, the MI terms are estimated with MINE Belghazi et al. (2018) based on a fully-connected neural network with one single hidden layer of dimension 100. The network is trained for 200 epochs and a batch size of 64, using the Adam optimizer with default parameters (on PyTorch).

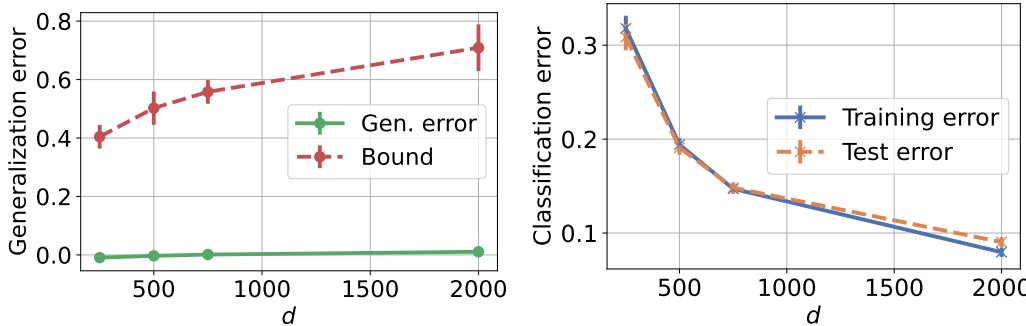

Figure 5: Generalization bounds on MNIST classification with neural networks trained on $\mathrm{W}_{\Theta,d}$

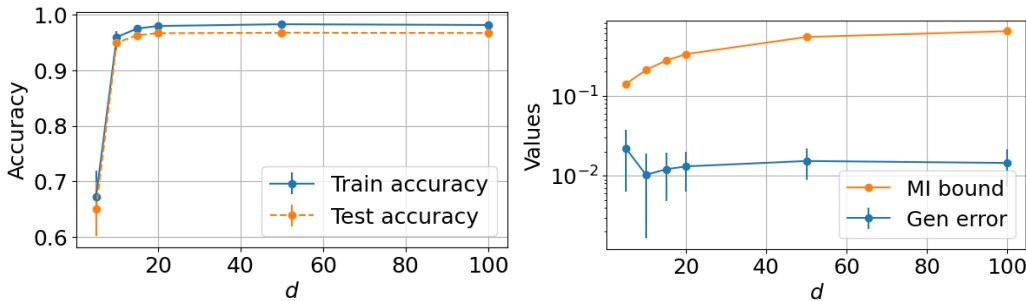

Figure 6: Generalization bounds on Iris dataset classification with neural networks trained on $\mathrm{W}_{\Theta,d}$

**Quantization method.** We use the quantization scheme of Lotfi et al. (2022), with minor modifications. We learn $c = [c_1, ..., c_L] \in \mathbb{R}^L$ quantization levels in 16-precision during training using the straight through estimator, and quantize the weights $W' = [W_1, \cdots, W_d] \in \mathbb{R}^d$ into $\widehat{W}_i = c_{q(i)}$, where $q(i) = \arg\min_{k \in \{1,...,L\}} |W_i - c_k|$. Post quantization, arithmetic coding is employed for further compression, to take into account the fact that quantization levels are not uniformly distributed in the quantized weights. Denote by $p_k$ the empirical probability of $c_k$. Arithmetic coding uses at most $\lceil d \times H(p) \rceil + 2$ bits, where $H(p) = -\sum_{k=1}^{L} p_k \log_2 p_k$. The total bit requirement for the quantized weights, the codebook $c$, and the probabilities $(p_1, \ldots, p_L)$ is bounded by $\lceil d \times H(p) \rceil + L \times (16 + \lceil \log_2 d \rceil) + 2$.

## B.2 ADDITIONAL DETAILS AND EMPIRICAL RESULTS FOR SECTION 4.1

**Binary classification with logistic regression.** We consider the binary classification problem solved with logistic regression as described in (Bu et al., 2019, §VI), with features dimension $s = 20$, hence $D = s+1$ (weights and intercept). We train our model on $\mathrm{W}_{\Theta,d}$ for different values of $d < D$, using $n$ training samples. We compute the test error on $\lfloor 20n/80 \rfloor$ observations. For each value of $n$ and $d$, we approximate the generalization error for 30 samples of $\Theta$ independently drawn from the SVD-based projector (see Section 4). We estimate the MI term in the bounds via MINE (with the aforementioned architecture) using 30 samples of $(W', Z_i) \sim P_{W'|S_n,\Theta} \otimes \mu$ for each $\Theta$.

**Multiclass classification with NNs.** We consider a fully-connected neural network with two hidden layers of width 200 to classify MNIST (LeCun & Cortes, 2010) and CIFAR-10 (Krizhevsky et al., 2009). The random projections are sampled using the Kronecker product projector, in order to scale better with the high-dimensionality of our models (see Appendix B.2). We train our NNs on $\mathrm{W}_{\Theta,d}$ for different values of $d$, including the intrinsic dimensions reported in Li et al. (2018). We approximate the generalization error for 30 samples of $\Theta$ and estimate our MI-based bounds given by Theorem A.2. The MI terms are estimated using MINE over 100 samples of $(W', Z_i) \sim P_{W'|S_n,\Theta} \otimes \mu$ for each $\Theta$. As MINE requires multiple runs, which can be very expen-

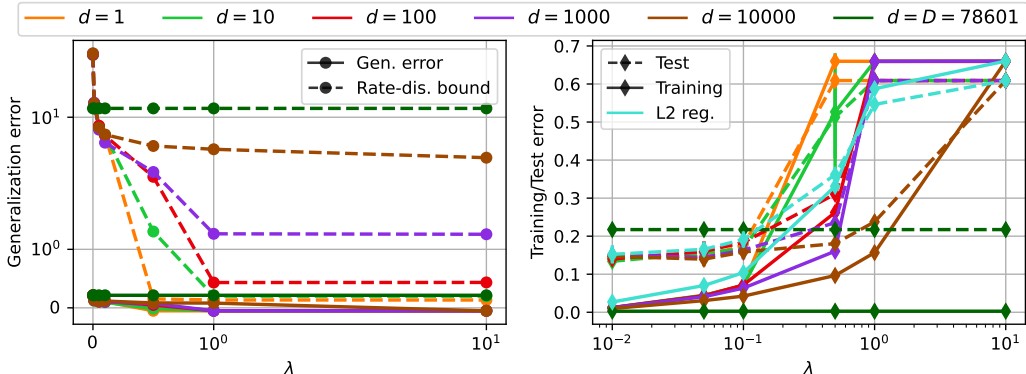

Figure 7: Influence of $(\lambda, d)$ on generalization errors and rate-distortion bounds *(left)*, and training and test errors *(right)* for a Lipschitz-constrained neural network on MNIST classification *in an "out-of-distribution" setting*. Results are averaged over 5 runs. The reported training/test error corresponds to the value of the loss (binary cross-entropy) on the training/test dataset.

sive, we only estimate MI for datasets and models of reasonable sizes: see Figure 5 for results on MNIST. For MNIST and CIFAR-10, we quantize $W'$ and evaluate our quantization-based generalization bounds. To train our NNs, we run Adam (Kingma & Ba, 2017) with default parameters for 30 epochs and batch size of 64 or 128.

We also classify the Iris dataset (Fisher, 1936). We train a two-hidden-layer NN with width 100 (resulting in $D = 10,903$ parameters) on $W_{\Theta,d}$. We use Adam with a learning rate of 0.1 as optimizer, for 200 epochs and batch size of 64. We approximate the generalization error for 20 samples of $\Theta$ independently drawn from the SVD-based projector. We evaluate our generalization bounds (Theorem 3.2) using MINE over 500 samples of $(W', Z_i) \sim P_{W'|\Theta,S_n} \otimes \mu$ for each $\Theta$. We report results for $d \in \{5, 10, 15, 20, 50, 100\}$ in Figure 6. We obtain over 95% accuracy at $d = 10$ already, and both the best train and test accuracy is achieved for $d = 50$. As expected, our bound is an increasing function of $d$ and all bounds are non-vacuous.

## B.3 Additional Details and Empirical Results for Section 4.2

**Lipschitz neural networks.** We follow the guidelines in Bethune et al. (2023) to design a Lipschitz-controlled neural network. We detail the computation of the Lipschitz constant of the neural network considered in Section 4.2. Denote $\hat{y} = f(w, x) = f_2 \circ \varphi \circ f_1(w, x)$. Then, the gradients of the loss with respect to $(w_2, b_2)$ are given by,

$$\nabla_{w_2} \ell(w, z) = \nabla_{\hat{y}} \mathcal{L}_{\text{CE}}(\hat{y}) \nabla_{w_2} (w_2[\varphi \circ f_1(w, x)] + b_2) \tag{90}$$
$$= \nabla_{\hat{y}} \mathcal{L}_{\text{CE}}(\hat{y})[\varphi \circ f_1(w, x)] \tag{91}$$

$$\nabla_{b_2} \ell(w, z) = \nabla_{\hat{y}} \mathcal{L}_{\text{CE}}(\hat{y}) \nabla_{b_2} (w_2[\varphi \circ f_1(w, x)] + b_2) \tag{92}$$
$$= \nabla_{\hat{y}} \mathcal{L}_{\text{CE}}(\hat{y}) \tag{93}$$

where $\mathcal{L}_{\text{CE}}$ denotes the binary cross-entropy.

Now, denoting $z'' = w_1 x + b_1$ and $z' = \varphi(z'')$,

$$\nabla_{w_1} \ell(w, z) = \nabla_{\hat{y}} \mathcal{L}_{\text{CE}}(\hat{y}) \nabla_{z'} (w_2 z' + b_2) \nabla_{z''} \varphi(z'') \nabla_{w_1} (w_1 x + b_1) \tag{94}$$
$$= \nabla_{\hat{y}} \mathcal{L}_{\text{CE}}(\hat{y}) w_2 \nabla_{z''} \varphi(z'') x \tag{95}$$

$$\nabla_{b_1} \ell(w, z) = \nabla_{\hat{y}} \mathcal{L}_{\text{CE}}(\hat{y}) \nabla_{z'} (w_2 z' + b_2) \nabla_{z''} \varphi(z'') \nabla_{b_1} (w_1 x + b_1) \tag{96}$$
$$= \nabla_{\hat{y}} \mathcal{L}_{\text{CE}}(\hat{y}) w_2 \nabla_{z''} \varphi(z'') \tag{97}$$

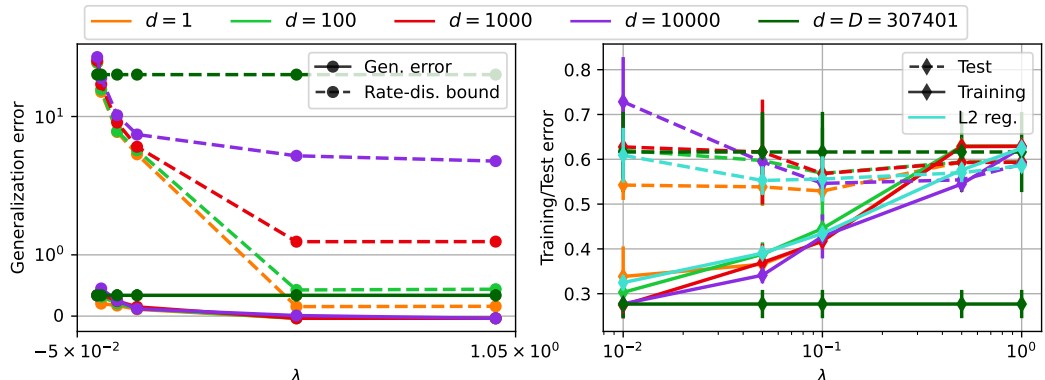

Figure 8: Influence of $(\lambda, d)$ on generalization errors and rate-distortion bounds *(left)*, and training and test errors *(right)* for a Lipschitz-constrained neural network on CIFAR-10 classification *in an "out-of-distribution" setting*. Results are averaged over 5 runs. The reported training/test error corresponds to the value of the loss (binary cross-entropy) on the training/test dataset.

Therefore,

$$\|\nabla_w \ell(w, z)\|^2 = \sum_{i=1}^{2} \left( \|\nabla_{w_i} \ell(w, z)\|^2 + \|\nabla_{b_i} \ell(w, z)\|^2 \right) \tag{98}$$

$$\leq \|\nabla_{\hat{y}} \mathcal{L}_{\text{CE}}(\hat{y})\|^2 \left\{ \|\varphi \circ f_1(w, x)\|^2 + 1 + \|w_2\|^2 \|\nabla_{z''} \varphi(z'')\|^2 [\|x\|^2 + 1] \right\} \tag{99}$$

$$\leq \|f_1(w, x)\|^2 + 1 + \|w_2\|^2 [\|x\|^2 + 1], \tag{100}$$

where (100) follows from the fact that the binary cross-entropy is 1-Lipschitz, and that ReLU activation is 1-Lipschitz and null at 0. By (100), we conclude that the Lipschitz constant of our loss (w.r.t. to $w$) is $L = \sqrt{(\|w_1\| \|x\| + \|b_1\|)^2 + 1 + \|w_2\|^2 [\|x\|^2 + 1]}$.

We normalize the data so that each pixel of any input $x \in \mathbb{R}^s$ lies between 0 and 1. The maximum norm of $x$ is then $\sqrt{s}$. We enforce the maximum norms of $b_i$ and $w_i$ to be $1/\sqrt{s}$ via projected gradient descent.

**Bounded loss.** If there exists $a \in \mathbb{R}$ such that $f(w, x) \geq a$, then the binary cross-entropy loss satisfies $\ell(w, z) \leq \log(1 + e^a)$. For $f(w, x)$ defined as the neural network above, one can show that $a = -M^2(\|x\|_2 + 1) + M$, where $M$ is s.t. $\|w_i\| \leq M$, $\|b_i\| \leq M$ for $i \in \{1, 2\}$.

**Additional Experiments.** We consider the same binary classification task with Lipschitz-controlled neural networks as in Section 4.2, but on a practical setting that is beyond the scope of our theory: we hide certain classes in the training set (e.g., digit 2) and include them in the test set. Given input image $X$, the label is $y = 1$ if $X$ represents a certain class (digit 1 for MNIST, 'automobile' on CIFAR-10), $y = 0$ otherwise. For MNIST, the training dataset contains digits $\{0, 1, 8\}$ and the test dataset contains digits $\{0, 1, 8, 2\}$; for CIFAR-10, the training dataset contains classes {automobile, cat, deer}, while the test dataset contains classes {automobile, cat, deer, truck}. Note that these classes were arbitrarily chosen. The generalization error and rate-distortion bounds are approximated over 5 $\Theta$ randomly drawn from the sparse projector. We use Adam for training, with default parameters, 30 epochs and batch size 128. For different $d$ and $\lambda$, we train on MNIST and CIFAR-10 with $n = 2000$ and approximate the generalization error and our rate-distortion bound given in Theorem 3.4 with $\delta = 1/n$: see Figures 7 and 8. Our empirical results shows that our discussion regarding the generalization ability of approximately-compressible NNs applies to this "out-of-distribution" setting as well: a more comprehensive investigation of generalization bounds on such settings is left for future work.

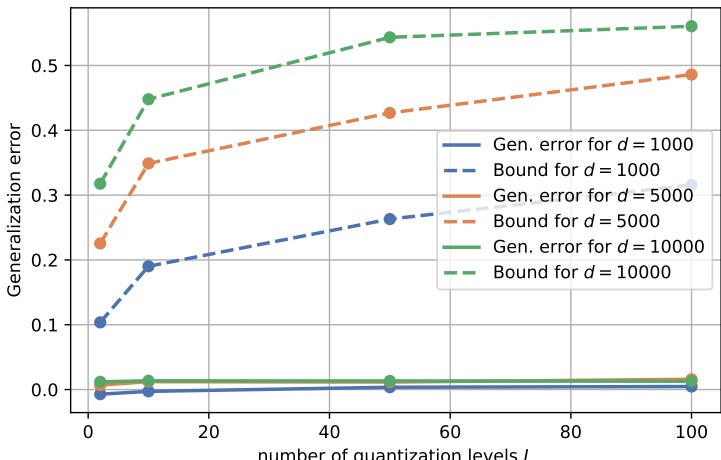

Figure 9: Influence of the number of quantization levels $L$ on the generalization error and our bounds, for MNIST classification with NNs.

## C    NEW EXPERIMENTS

**Influence of the number of quantization levels.** We analyze the influence of the quantization levels $L$ on the generalization error and our bounds in practice. We consider the MNIST classification with NNs (described in Section 4) and train for varying $L$. We report the results in Figure 9 for different values of $d$. We observe that for all tested dimensions, the generalization error increase with increasing $L$. Our bound exhibits the same behavior, which we expected given their dependence on $L$ (see paragraph "Quantization" in Section 4). This experiment illustrates that *(i)* the more aggressive the compression, the better the generalization, *(ii)* our bounds accurately reflect the behavior of the generalization error, and seem tighter for lower values of $d$ and $L$.

**Rate-distortion bounds.** We evaluate our rate-distortion bounds on binary classification using the same neural network as in Appendix B.3. The training and test data consists of MNIST images corresponding to digits 1 and 7. To make generalization harder, we train our model on $n = 10$ samples. The maximum norm of the weights and biases is constrained to 1 via projected gradient descent. The optimization procedure is the same as in Appendix B.3 and the results are reported in Figure 4. Note that the bounds are overall looser than in Figure 7: this is due to the fact that we chose a larger value for the maximum norm of $w_i, b_i$, thus the Lipschitz constant $L$ is larger.

