# OpenReview forum: "Slicing Mutual Information Generalization Bounds for Neural Networks"
_ICLR.cc/2024/Conference — Submitted to ICLR 2024_

### Official Review · Reviewer_JYMx · 2023-10-28

**Soundness:** 3 good
**Presentation:** 3 good
**Contribution:** 2 fair
**Rating:** 5
**Confidence:** 5

**Summary:**

This paper establishes information-theoretic generalization bounds for learning algorithms trained on random subspaces. By assuming that the D-dimensional model weights w can be projected into a d-dimensional subspace w’ by w = Theta w’, the authors present tightened versions of generalization bounds in (Xu et al, 2017) and (Bu et al, 2019) by replacing w with w’. The authors further connect these results with sliced mutual information under some simple learning scenarios, and present rate-distortion bounds by incorporating the Lipschitz assumption.

**Strengths:**

1. The presented bounds effectively lower the dimensionality of random variables used in the key mutual information terms, improving the computational tractability of these bounds.
2. The established bounds successfully reflect the trend of the true generalization gap, and are shown to be tighter than previous competitors.

**Weaknesses:**

1. The target learning scenarios seems too restrictive for modern deep-learning models. The subspace mechanism severely restricts the available number of parameters in the neural network, making them hardly applicable in modern learning tasks. The Lipschitz condition is also hardly satisfied in conventional network architectures. As seen in Figure 7, the training accuracy cannot reach 80% on CIFAR10, even for a binary classification task.

2. Although the dimensionality is reduced in the presented bounds, this improvement hardly solves the tractability problem. The key mutual information I(W’;S) is still computationally intractable for greater d values, e.g. d > 20. For empirical analysis, the authors adopt a neural network-based mutual information estimator (MINE), whose accuracy lacks theoretical guarantees and is thus questionable for high-dimensional variables. The viability of applying MINE for theoretical analysis is doubtful.

3. The derivation of Theorem 3.1 and 3.2 seems trivial for me: given a fixed Theta, the trainable set of parameters then becomes W’ instead of W. Then Theorem 3.1 and 3.2 directly follows by Xu and Bu’s results by replacing I(W;S) with I(W’;S), or replacing I(W;Z_i) with I(W’;Z_i). The current contribution of the paper may not be sufficient to be published in ICLR.

**Questions:**

Please refer to weaknesses

---

> ### Author Response · Authors · 2023-11-17
> **Response to Reviewer JYMx**
>
> We thank the reviewer for their comments. We hope our answers below address their concerns, and would be happy to discuss if they have further questions.
>
> **Answers to questions:**
>
> 1. It seems like there was some unfortunate confusion regarding Figure 7: the $y$-axis reports the error as measured by the binary cross-entropy loss on the test set. The corresponding training accuracy can reach 100\% after 100 epochs (for $\lambda=0$). Note that the size of the training dataset is $2000$ to make generalization harder. We make this clearer in the revision. The remaining concerns raised are fortunately not an obstacle. First, the random compression scheme we study has found tangible success in prior recent work: we refer the reviewer to our general answer for detailed explanations. Second, regarding *"the Lipschitz condition is hardly satisfied in conventional network architectures"*: feedforward NNs based on common choices of activations (e.g., ReLU) can easily be made Lipschitz with respect to the weights by simply constraining the maximum norm of the data input and weights (Bethune et al., 2023). We also emphasize that the Lipschitz condition is a common strategy to control generalization error bounds that contain a distortion term (Sefidgaran et al., 2022; Neyshabur et al., 2017: see our response to Reviewer qRo1).
> &nbsp;
> 2. *"The viability of applying MINE for theoretical analysis is doubtful"*: On the contrary, theoretical guarantees on the convergence rate of MINE are well studied and strong, see e.g. Theorem 2 of Goldfeld et al. (2022) which studies MINE in the context of random slicing and shows a clear error bound decaying as the dimension $k$ of the slices (our $d$) decrease. We will add a remark in the revision pointing out this fact. Also, see Remark 2 in Goldfeld et al. (2022) for further discussion of relaxation of some of the assumptions of MINE, which enables optimizing over a larger class of distribution in lower-dimensional spaces. While  the results in Goldfeld et al. (2022) are stated for 3 layer MINE networks for simplicity, the extension to deep networks is given in followup works if the reviewer is interested. Note the regime discussed by McAllester & Stratos (2020) (see the discussion in our Introduction) is not relevant for us since it applies to the case of large MI -- in that setting, the generalization error bound would be very large and vacuous anyway. We added these elements in our paper to clarify why and when our approach is beneficial for estimating MI.
> &nbsp;
> 3. While our paper has an important theoretical component, the main motivation and message of our paper are practical (cf. our general response to all reviewers). Given the growing interest for NNs trained on random subspaces (Li et al., 2018; Lotfi et al., 2022), our work seeks to derive insights and build theoretical foundations on their generalization ability. We do not need our proof techniques to be difficult or of independent interest to achieve this goal. In fact, we would argue there is strength in simplicity. That said, we emphasize that our theoretical results in Section 3.2, which result from applying our generic bounds in specific settings, are nontrivial. Even for Gaussian mean estimation and linear regression, the derivation of the generalization error and the bound on its cumulant-generating function is not straightforward, e.g. the loss follows a more complex distribution (generalizated chi-square) than in the original space (Sections A.3 and A.4).

---

> > ### Author Response · Authors · 2023-11-20
> >
> > Dear Reviewer JYMx: Thank you again for your time and your feedback, which have helped us improve our paper. We hope our responses have adequately addressed your questions, and that you will reconsider your score in light of our discussion. As the discussion period ends soon, we would greatly appreciate it if you could let us know any remaining questions or concerns you may have: we are more than willing to provide additional explanations and experiments to clarify these.

---

> > ### Comment · Reviewer_JYMx · 2023-11-21
> > **Reply to author's response**
> >
> > Thank you for the response. After the rebuttal, I am still concerned about the adoption of MINE. It seems that the error bound provided by Goldfeld et al. (2022) easily becomes vacuous for larger $d$, as it grows with $d^{3/2}$. The claim that the generalization bounds derived are computationally friendly seems to be too strong. The capacity of the compressed models is not clear, as only the test risk is reported, not the test accuracy. I remain doubtful about how well these models will perform on more complex learning tasks. Meanwhile, the theoretical contribution is a bit limited, and the results are not surprising: restricting the hypothesis space of course leads to better generalization. Base on that,  I would like to keep my current rating.

---

> ### Author Response · Authors · 2023-11-21
> **MINE**
>
> Indeed, the theoretical bound grows with $d$, but recall this is in no way a problem, since the $d$ here is the small dimension of the slice, not the ambient dimension. Furthermore, observe that $d^{3/2}$ is only slightly worse than linear scaling.  We reiterate that MINE is a well-established, empirically highly successful estimator even in very high dimensions; and should not be a cause for concern in this relatively low-dimensional estimation regime. Indeed our empirical results confirm it can be used successfully here - not one of our generalization bounds computed in this way is overly optimistic.
>
> "Restricting the hypothesis space of course leads to better generalization." Agreed, this sums up the field of Learning Theory - yet the field is considered one of the most difficult in theoretical computer science. The challenge, unfortunately, is always going to be identifying a restricted hypothesis space that is useful in practice. Random slicing, we argue, is such a mechanism - in fact one highly amenable to the setting of practically computable information-theoretic generalization bounds. As such, we strongly believe that it will be of interest to the community.

---

### Official Review · Reviewer_KfBv · 2023-10-30

**Soundness:** 2 fair
**Presentation:** 3 good
**Contribution:** 2 fair
**Rating:** 6
**Confidence:** 3

**Summary:**

In this paper, the authors build upon "mutual information generalization bound" framework, and extend it by studying a setting with:
* parameters living in low-rank subspace, or close to it
* weight quantization
* Lipschitz losses, bounded weights

They derive new bounds showing that, if the weight are optimized over a low-dimensional manifold, tighter bounds can be obtained than the one of literature. They also propose a new "rate distortion" bound that relies on the Lipschitz constant of the loss.

Moreover, authors characterize the behavior of their bounds in the following cases:
* Gaussian mean estimation
* Linear regression

Showing that their method effectively extends the work of Xu & Raginsky (2017), as well as Bu et al. (2019).

Finally, authors measure empirically their bound on high dimensional problem, e.g. training of a deep neural network on Mnist and Cifar-10. Since the bounds rely on mutual information, they use mutual information estimators like MINE to evaluate the bound.

**Strengths:**

### Originality

The work is original, improving upon previous work to take into account the dimension of the parameter manifold, and the importance of quantization.

### Quality

Both toy models, and deep neural network training is studied. Insight can be useful for practionners and theoreticians. Experiments are convincing.

### Clarity

The paper is clear overall, but not without defaults (see comments below). The literature review is very accessible. The authors do a good job at making the theorem A.2 more accessible with theorems 3.1 and 3.2.

### Significance

Mutual information bounds are useful, and can be evaluated empirically with MIN estimator. This work gives an additional motivation to study and optimize low-rank neural networks. Code is given.

**Weaknesses:**

### Clarity

There is a lack of clarity or details at times (see **Questions**).

### Bias-variance tradeoff

>  On the other hand, decreasing d may increase the training error, implying a tradeoff between generalization error and training error when selecting d.

and

>  The choice
of d is also important and can be tuned to balance the MI term with the distortion required (how
small λ needs to be) to achieve low training error.

Most of the bounds derived by the author can be understood as a bias-variance tradeoff, with somes terms that decrease with $n$ (the *variance*), and some other that do not decrease with $n$ (the *bias*). Thereore, it is a bit upsetting to not see a clear discussion of this. For example, in the rate-distortions bounds (theorems 3.3 and 3.4) the left term does not depend on $n$. What are the practical implications of this?

Maybe it is worth adding a discussion on this topic, and to underline the terms of your bounds with `\underbrace`.

### Quantization

Theoretical results show a dependency of the bound on the number of levels in quantization. However, in none of the experiment the number of levels of quantization is kept constant in experiment of Fig 3. It would be interesting to measure the generation bounds when the number of level vary, compared to the (empirical) generalization error.

### Motivation behind the orthogonal projector

The motivation behind introducing $\Theta$ is unclear to me. It *seems* that sampling an element from the Grassmannian $\mathcal{M}$ would be enough: https://en.wikipedia.org/wiki/Grassmannian It seems that it is not crucial that $w=\Theta w'$ for the theoretical results to hold. What *seems* to matter is that $w$ is effectively low-rank, i.e. live on a low-dimensional manifold $\mathcal{M}$ which is sampled independantly from the data, such that $\mathbb{P}(\mathcal{M}|Z)=\mathbb{P}(\mathcal{M})$ so that expectation can be taken over $\mathbb{P}(\mathcal{M})$ in the bound. The fact that $\Theta$ is orthogonal does not seem to be necessary as long as $w$ is sampled from $\mathcal{M}$.

Is there a motivation behind using orthogonal matrices $\Theta$ ? Getting rid of it would allow to get rid of the `scipy.linalg.orth` in the experiments.

### Experimental setup of sec 4.2 doesn't match theorems

The theorems assume that the train set $S_n$ is sampled from $\mu$. However, as written in appendix:

> To make generalization harder, we hide a class during training. Specifically, for MNIST, the training dataset contains digits {0, 1, 8} and the test dataset contains digits {0, 1, 8, 2}; for CIFAR-10, the training dataset contains classes {automobile, cat, deer}, while the test dataset contains classes {automobile, cat, deer, truck}.

This makes the test set Out Of Distribution (OOD) compared to the test. It is not Empirical Risk Minimization (ERM) paradigm anymore, but rather evaluation in face of  distribution shift. Therefore, the generalization error reported cannot be compliant with the bound. For example, the marginal probabilities of the $y=1$ class falls from $\frac{1}{4}$ to $\frac{1}{5}$ for Mnist, and from $\frac{1}{3}$ to $\frac{1}{4}$ for Cifar-10.

I am not convinced about the relevance of this setting. Sticking with ERM paradigm would be better.

### Typo: arguments switched

* In page 1 the loss is defined as $\ell:Z\times W\rightarrow\mathbb{R}$, but everywhere else it is used as $\ell(w,z_i)$.

**Questions:**

### Bounded binary cross entropy

You write (p9):
> The role of $\epsilon>0$ is to make $\ell$ bounded thus meet the conditions of Theorems 3.3 and 3.4.

I failed to be convinced. If no constraint is put on $f(w,X)$ then $f(w,X)+\epsilon$ can take any value in $(-\infty,+\infty)$ range. Unless $f$ is lower bounded. When reading the appendix (p21) that seems to be the case since $w_i$, $b_i$ and $x$ are bounded. Unfortunately, the bound on $f$ is not given, and the link with boundedness of $f$ is not explicitly drawn. Can you clarify?

### Unbounded MI

>  for instance if $W ′$ is a deterministic function of $S_n$ given $\Theta$ then $I_{\Theta}(W ′; S_n) = +\infty$,

Since Mutual Information is bounded by entropy of variables involved: $I_{\Theta}(W ′; S_n)\leq H(S_n)$, do you mean that $H(S_n)$ is unbounbed?

### Appendix

Can you clarify how you went from step (39) to step (40) in appendix, p15 ? Idem for steps (42) to (43).

### Conclusion

I would be happy to raise my score upon satisfying answer to the questions and remarks.

---

> ### Author Response · Authors · 2023-11-16
> **Response to Reviewer KfBv**
>
> We thank the reviewer for their feedback: we are glad they found our work "original" and "useful", our experiments "convincing", and our paper "clear overall".
>
> **Bias-variance tradeoff:** This is an insightful connection to make, indeed the bias-variance tradeoff is spiritually very similar to the rate-distortion tradeoff. That said, the concept of rate-distortion is the more applicable concept here since we are using the framework of compression and information content. As noted, the first term will be the distortion (c.f. bias) which will decrease as $d$ increases, but will not go away with $n$. The second term is the rate (c.f. variance) which will decay with both $n$ and $d$. We will add in a more extended discussion of this tradeoff and how it guides the choice of $d$; we initially had to omit it for space reasons.
>
> **Quantization:** As suggested by the reviewer, we added an additional experiment to illustrate how the number of quantization levels influences the empirical generalization error and our bound. Our results and interpretation are provided in Section C, and confirm our theoretical analysis.
>
> **Motivation behind the orthogonal projector:** The reviewer's intuition makes sense: our theoretical results could be extended to more general low-rank representations of $W$. We actually stated Lemma A.1 so that it is generic enough (i.e., not restricted to $w = \Theta w'$) to facilitate such extensions.
> That being said, we believe it is important to consider orthonormal projectors $\Theta$ for several reasons. First, prior work showed that this is an interesting setting in deep learning, e.g., to efficiently compress NNs while maintaining a reasonable performance, or to tighten PAC-Bayesian generalization bounds: see paragraph "Sliced neural networks" in Section 1 or our paper. Then, assuming $\Theta^\top \Theta = I_d$ simplifies some derivations when computing the generalization error and its bound, e.g., for Gaussian mean estimation (Section A.3). Last but not least, orthonormality is advantageous for computational purposes, as it prevents poor conditioning when training on $\mathrm{W}_{\Theta, d}$ (Lotfi et al., 2022).
>
> **Experimental setup of sec 4.2:** This is a fair point: we added the experiment suggested by the reviewer (see Section C). Instead of hiding a class in the training dataset to make generalization harder, we reduced the size of the dataset. Our plot further illustrates our rate-distortion bounds and is line with our intuition that approximately-compressible NNs (i.e., with smaller distortion term) generalize better.
>
> **Typo:** The typo is now fixed in our revision.
>
> **Bounded binary cross entropy:** We indeed forgot to explicitly mention that $f(w,X)$ needs to be lower-bounded for $\ell$ (the binary cross-entropy) to be bounded -- and as correctly pointed out by the reviewer, this is the case in our experiments (Sections 4.2 and B.3). The hyperparameter $\varepsilon$ can thus be used as a way to adjust the bound on $\ell$ hence the generalization bound, if desired. We thank the reviewer for raising this point and we clarified it in our revised paper: see Sections 4.2 and B.3.
>
> **Unbounded MI:** The inequality $I^\Theta(W';S_n) \leq H(S_n)$ does not hold in our general setting, where $S_n$ is a continuous random variable. By eq.(3), we can only say that $I^\Theta(W';S_n)=h^\Theta(S_n)-h^\Theta(S_n|W')$, where $h^\Theta(S_n)\triangleq - \int p(x|\Theta'=\Theta) \log p(x|\Theta'=\Theta) \mathrm{d}x$ and $h^\Theta(S_n|W') \triangleq - \int p(s_n|w', \Theta'=\Theta) \log p(s_n,w'|\Theta'=\Theta) \mathrm{d}s_n \mathrm{d}w'$. Since $p(s_n|w', \Theta'=\Theta)$ denotes a probability density function, it can take values greater than 1, thus $h^\Theta(S_n | W')$ can be negative. Additionally, $I(W';S_n)=+\infty$ if $W'=g(X)$ with $g$ a smooth, non-constant and deterministic function. We clarified this point in Section 3.1 of our revised paper.
>
> **Appendix:** We added more explanations on *"step (39) to step (40)"* and *"step (42) to step (43)"*: see Section A.3.

---

> > ### Comment · Reviewer_KfBv · 2023-11-17
> > **Thank you for your clarifications**
> >
> > I am satisfied with your answer regarding **Bias-variance tradeoff**, **quantization**, **Motivation behind the orthogonal projector** and the other clarifications. I still have a remaining concern:
> >
> > **Experimental setup of sec 4.2:** I think this limitation (the distribution shift of the test set) should be clearly specified in the caption of Figure 4, since you decided to keep it in the main text. I noticed that Figure 4 is not referenced in Section 4.2 (nor anywhere else), so that may be a typo too. Fig 9 would convey the message of your paper better, in place of Figure 4.

---

> > > ### Author Response · Authors · 2023-11-17
> > > **Thank you**
> > >
> > > Thank you for your response and for increasing your original rating: we are glad that you are satisfied with our clarifications!
> > >
> > > **Experimental setup of sec 4.2:** Thank you for your additional feedback: we have fixed the typos, moved Figure 9 in the main text of the revised paper, and moved Figure 4 in the supplementary document (see Section B). We hope this adequately addresses your remaining concern.

---

### Official Review · Reviewer_qRo1 · 2023-11-01

**Soundness:** 3 good
**Presentation:** 3 good
**Contribution:** 3 good
**Rating:** 6
**Confidence:** 4

**Summary:**

This paper presents generalization bounds for a learning algorithm, utilizing an information-theoretic measure of dependence between the output of the learning algorithm and the training dataset. The primary motivation behind this work is to address the statistical challenges associated with estimating generic information-theoretic generalization bounds.

The paper introduces the concept of "slicing" the network's parameters and provides bounds for learning algorithms that focus on updating only a specific "slice" of parameters. The primary advantage of this approach lies in the ease of estimating mutual information-based bounds.

**Strengths:**

This paper attacks an important problem regarding obtaining an information-theoretic generalization bound that is easy to estimate. The idea based on the slicing seems very interesting and it gives rise to a connection to the sliced mutual information.

**Weaknesses:**

My main concern with the paper is that the main message of the paper is not clear. I think the idea of projection of parameters is interesting and it is intuitive that we can have a smaller generalization error. My understanding of the main message of this paper is that: slicing is interesting since we can have a better estimator of the information-theoretic generalization bounds. However, I think that only obtaining numerical values may not the only goal of the generalization theory.  I appreciate if the authors provide more discussion regarding the main message of this paper.

Also, regarding the idea of compressibility, it is not clear why a "data-independent" projection matrix will be able to find the best subspace.

**Questions:**

1- Motivation of this paper: Could you please provide a more detailed discussion on the motivation behind your paper, addressing the points raised in the weaknesses section?

2- Intrinsic Dimension: Throughout your paper, there is a recurring mention of the intrinsic dimension. Could you clarify what you mean by "intrinsic dimension"?

3- Comparison between your work and "functional" or "evaluated" MI bounds: In the existing literature, there are generalization bounds that do not suffer from the curse of dimensionality. It would be valuable to understand how your proposed bounds in this paper compare to the results presented in the following papers:

-- Harutyunyan, Hrayr, et al. "Information-theoretic generalization bounds for black-box learning algorithms." Advances in Neural Information Processing Systems 34 (2021): 24670-24682.

-- Haghifam, Mahdi, et al. "Understanding generalization via leave-one-out conditional mutual information." 2022 IEEE International Symposium on Information Theory (ISIT). IEEE, 2022.

4- Different Compression Methods: Your paper discusses modifying learning algorithms to improve information-theoretic generalization bounds. This approach reminds me of adding Gaussian noise to the output of a learning algorithm and trading off information with distortion, which has been explored in other works, such as:

-- Neyshabur, Behnam, Srinadh Bhojanapalli, and Nathan Srebro. "A pac-bayesian approach to spectrally-normalized margin bounds for neural networks." arXiv preprint arXiv:1707.09564 (2017).

-- Dziugaite, Gintare Karolina, and Daniel M. Roy. "Computing nonvacuous generalization bounds for deep (stochastic) neural networks with many more parameters than training data." arXiv preprint arXiv:1703.11008 (2017).

Could you outline the main differences between your approach and the approach based on adding Gaussian noise?

5- PostHoc explanation of low dimensional compressibility: This paper is motivated by the observation that many neural networks are highly compressible. However, it seems that the method based on random projection might not capture low-dimensional compressibility adequately since the subspace might be data-dependent. Additionally, one limitation of your paper is that it's not clear how to choose the dimension for projection (d). Could you provide insights or potential solutions to address these concerns?

6- Connections to Pruning Methods:  Can you explain how your findings are connected to pruning techniques used in neural networks?

---

> ### Author Response · Authors · 2023-11-16
> **Response to Reviewer qRo1**
>
> We thank the reviewer for their positive evaluation of our paper. We appreciate they find our idea "very interesting". Please find our replies below.
>
> **Answers to questions:**
>
> 1- **Motivation.** We provided a clearer picture of the motivations and key contributions of our work: please see our general answer to all reviewers. We hope our additional explanations addresses the reviewer's question, and we are happy to further clarify any aspect during the discussion period.
>
> 2- **Intrinsic Dimension.** We refer by "intrinsic dimension" the notion described in Section 2 of (Li et al., 2018), since our setting is largely inspired by this work. We added this clarification in the revised manuscript to resolve any ambiguity. Briefly, the "intrinsic dimension" is the slice dimension $d$ below which the test performance of the sliced neural network begins to degrade significantly.
>
> 3- **Comparison between your work and "functional" or "evaluated" MI bounds.** We thank the reviewer for sharing these two references: a discussion on how their strategies compare to our work is provided in Section 2, paragraph "Conditional MI generalization bounds". We will add the second reference (Hafigham et al., 2022) to our manuscript (the first one, Harutyunyan et al., 2021, was already cited). We will be happy to complement this discussion to address any questions the reviewer may still have.
>
> 4- **Different Compression Methods.** The two references provided by the reviewer use an approach that is in essence different from ours: generalization error bounds for NNs are derived by *adding a random perturbation to the weights* and using *PAC-Bayes theory*, whereas we *project the weights onto lower-dimensional random subspaces* and use *information theory*. We view these two compression approaches to be orthogonal yet complementary, in the sense that the two compression methods address two very different intuitions about the structure of neural network weights (low rank structure vs. smoothness), but could easily be combined in practice (this would be an interesting direction for future work). One similarity with our work is that neural networks are *Lipschitz w.r.t the weights* to *bound a distortion term*: Lemma 2 in Neyshabur et al. (2017) applies to NNs with ReLU activations and weights with bounded norms; we implemented the same NNs to illustrate our rate-distortion bounds, since these are Lipschitz w.r.t the weights, as we show in our Section B.3. We will add this discussion in our paper.
>
> 5- **PostHoc explanation of low dimensional compressibility.** The use of random projections is a simple approach to (a) check in practice if neural networks are highly compressible (this is often the case as revealed by prior work, e.g., Li et al., 2018), (b) understand the impact of the subspace dimension on the generalization capacity (this is our contribution). In fact, learning $\Theta$ in a data-dependent manner would entirely defeat the purpose of our approach, as these information-theoretic generalization bounds require the accounting for any parameters that are data-dependent. The empirical success of random $\Theta$ was what inspired us to write this paper, as the fact we can ignore a random $\Theta$ in the mutual information computation is a massive savings.
>
> That said, we admit that one limitation of this method is the need to find the intrinsic dimension: the common method to do so is by trial-and-error (Li et al., 2018; Lotfi et al., 2022). Our generalization bounds improve on this situation, as our rate-distortion bounds in particular provide a generalization bound that can be optimized in terms of $d$, trading off the cost of larger $d$ and the benefit of lower distortion. To speed up the search of the intrinsic dimension, one can run the training for different $d$ in parallel, or use search algorithms with favorable time complexity.
>
> 6- **Connections to Pruning Methods.** This is an interesting question. First, we note that training on $W_{\Theta, d}$ is a more effective and flexible method for model compression than pruning, since $\Theta$ can be dense (see Lotfi et al., 2022). Then, while our theorems hold for any distribution $P_\Theta$ on the Stiefel manifold, we believe our proof techniques could be adapted to study more general low-rank representations of the parameter space (as implied by our Lemma A.1 and pointed out by reviewer KfBv). In that sense, our findings can be extended to gain more insights on the generalization ability of neural networks compressed via *random pruning*. However, pruning methods that remove weights in a *data-dependent manner* (such as magnitude-based pruning) do not fit our framework, since only data-independent dimension reduction approaches simplify the mutual information bounds.

---

> > ### Author Response · Authors · 2023-11-20
> >
> > Dear Reviewer qRo1: Thank you again for your time and your feedback, which have helped us improve our paper. We hope our responses have adequately addressed your questions, and that you will reconsider your score in light of our discussion. As the discussion period ends soon, we would greatly appreciate it if you could let us know any remaining questions or concerns you may have: we are more than willing to provide additional explanations and experiments to clarify these.

---

> > > ### Comment · Reviewer_qRo1 · 2023-11-22
> > > **Thanks**
> > >
> > > Regarding the difference between sliced MI and evaluated or functional CMI, my understanding from Harutyunyan et al., 2021 paper is that the goal in that paper is to devise an approach for "efficient" estimation of MI bounds, and the numerical results in Harutyunyan et al., 2021 seem tight. To me, this fact has not been discussed in the prior work section.
> > >
> > > Other than that the response answers my questions.

---

> > > > ### Author Response · Authors · 2023-11-22
> > > >
> > > > Thank you for getting back to us: we are glad our response addressed your questions.
> > > >
> > > > As you suggested, we added some explanations in Section 2 to clarify the comparison with Harutyunyan et al., 2021: **see Section 2 (changes are in blue)**. Briefly, their generalization bounds depend on a mutual information term evaluated on a prediction function rather than the weights. Since this predictor has a lower dimension, their bounds can indeed be evaluated efficiently. However, as they rely on the CMI approach, they require two training datasets, which could have been used to obtain a better estimate of the generalization error directly. In addition, it is unclear how such bounds could be interpreted for selecting model architectures; in contrast, our bounds clearly reflect the impact of practical compression schemes on generalization.

---

### Official Review · Reviewer_X27N · 2023-11-06

**Soundness:** 3 good
**Presentation:** 2 fair
**Contribution:** 2 fair
**Rating:** 5
**Confidence:** 4

**Summary:**

This paper presents an information-theoretic generalization bound for algorithms trained on random subspaces. This approach is driven by the challenge of estimating the input-output mutual information (MI) generalization bound for modern neural networks, which is hindered by the high dimensionality of the parameters. Furthermore, the paper introduces a training procedure based on a rate-distortion framework that allows computation of the MI bound using less restrictive weights, all the while preserving performance.

The paper makes the following contributions: (1) The authors proposed a novel information-theoretic generalization bound for algorithms trained on random subspaces (Section 3.1). (2) The authors showed the connection between this bound and  k-SMI in two settings: Gaussian mean estimation and linear regression (Section 3.2). (3) The authors showed empirically that their bounds are tighter compared to the bounds in Bu et al. (2019) (Section 4.1). (4) The authors proposed a training algorithm that allows computation of the information-theoretic generalization bound on neural networks with less restricted weights. This is done by introducing a regularization method that encourages the weights to be close to the random subspace. They showed that this approach improves both the generalization bound and test performance (Section 4.2).

**Strengths:**

The paper explores tightening information-theoretic generalization bounds while introducing a technique to evaluate this bound in high dimensional settings. This is indeed an interesting and important topic in deep learning. The proposed training procedures based on the rate-distortion framework is also novel and interesting.

**Weaknesses:**

The sliced information-theoretic bound, while interesting, may be computationally expensive especially in high dimensions. The bound requires training multiple models for multiple projection matrices. The authors resolved this by using quantization that avoids the estimation of MI, but no comparison was made on how loose this bound is compared to the sliced MI bound. Overall, the bounds also become increasingly loose as the dimension increases. Also, more evaluation is needed for Section 4.2 to demonstrate the effectiveness of this approach.

**Questions:**

1. In the footnote 3, the authors mentioned that their bound can be shown theoretically to be tighter for the two settings (mean estimation and linear regression) using data processing inequality. Can this detailed discussion be included and made clearer in the main text? Also, can it be shown theoretically that the bound is tighter in a more general setting (e.g., with neural networks)?

1. Is it possible to conduct empirical evaluation of the linear regression generalization bound?

1. Is it possible to empirically evaluate how varying the number of projection matrices and number of samples of $(W’,Z_i)$ affects the bound?

1. For Section 4.1, the authors evaluate the bound using the quantized weights and bypass the estimation of MI by considering the upper bound on $I^{\Theta}(W’;S_n)$. Is it possible to conduct a similar experiment to evaluate how loose this upper bound is compared to the bound in Theorem 3.1?

1. For Section 4.2, is it possible to vary the generalization error by changing the number of samples (similar to Figure 2), and evaluate the rate-distortion bound accordingly?

1. The writing and details provided in the paper could be improved to make the paper more readable and useful. For example: the paper contains a multitude of bounds but does not provide a unifying perspective. It would be good to have more discussion on the bounds and when to apply each one.

1. Overall, the bounds become increasingly loose as the dimension increases. This limits the value and applicability of the bounds in general. Is it possible to tighten the bounds and say something about converse bounds?

---

> ### Author Response · Authors · 2023-11-16
> **Response to Reviewer X27N**
>
> We thank the reviewer for finding our contributions "novel" and "interesting", and we appreciate they find the topic "important". We address their questions below. We also refer the reviewer to our general answer for a better picture of the main story and impact of our work.
>
> **Answers to questions**
>
> 1. In Gaussian mean estimation, $W' = \Theta W$ where $W$ is the solution of the original problem (the weights are optimized on $\mathbb{R}^D$). By the data processing inequality, $I(W';Z_i) \leq I(W;Z_i)$. Therefore, we obtain tighter bounds as compared to strategies which ignore the existence of an intrinsic dimension. The same reasoning applies to linear regression, since $W' = \phi(\Theta,X)W$. In general settings such as neural networks, the question remains open: the relation between $W'$ and $W$ is unknown, thus we cannot conclude using data processing inequality. We added this discussion in Section 3.2 to clarify footnote 3.
> &nbsp;
> 2. The empirical evaluation of the linear regression generalization bound requires finding the infimum in eq. (69). The solution is not available in closed form and can be approximated in practice using numerical solvers.
> &nbsp;
> 3. The generalization error and our bounds decrease with increasing number of training samples: see Figures 1 and 2 for illustration. Increasing the number of samples $(W', Z_i)$ will improve the accuracy of the estimates of $I^\Theta(W';Z_i)$ given by MINE. Finally, increasing the number of random projections for a fixed $d$ has little practical impact on our bounds. This is consistent with prior work (Li et al, 2018), which showed that the test performance does not vary much accross multiple projection matrices for a fixed $d$, while the choice of $d$ has a greater impact on the quality of solutions. We will add empirical results illustrating our conclusions.
> &nbsp;
> 4. We evaluated our generalization bounds for projected weights (Figure 5) as well as projected **and** quantized weights (Figure 3) for MNIST classification with NNs. We also ran an additional experiment in Section C. Interestingly, we see that for appropriate choices of $L$, the bounds for projected and quantized weights are tighter than when only projecting.
> &nbsp;
> 5. We provided more insights on the impact of $n$ on the rate-distortion bound: see paragraph "Bias-variance tradeoff" in our answer to Reviewer KfBv. We will include the experiment suggested by the reviewer.
> &nbsp;
> 6. The key message of our paper is the observation that using random compression tightens information-theoretic generalization bounds for neural networks. As suggested by the reviewer, we added a paragraph in Section 3.1 explaining when to apply each of our bounds. We hope this clarifies our contributions and will be happy to take into account further feedback on that point.
> &nbsp;
> 7. The reviewer is correct in saying that our bounds "become increasingly [large] as the dimension increases", but this is precisely an important take-home message of our work: **reducing the dimension of the training space (through random projections) tightens information-theoretic bounds on the generalization error**. This feature is then leveraged to produce tighter generalization bounds for *approximately-compressible NNs*. In that sense, we disagree with *"this limits the value and applicability of the bounds"*. In fact, it provides a theoretical confirmation of the practitioner-driven observed empirical success of random projection, and is in that sense certainly relevant practically. Indeed, our rate distortion bounds motivated a further modification of the random projection training procedure, allowing for increased flexibility. While a full exploration of the empirical properties of that algorithm is beyond the scope of this first paper, the initial results we show are encouraging. Finally, generalization bounds themselves do have practical utility, in that they can be used to guide difficult architecture choices and in the future inform "scaling laws" (e.g. in our case, the tradeoff between random projection and distortion). Scaling laws generally are often used practically, e.g. the Chinchilla scaling laws which enabled much of the recent growth in transformer architectures for LLMs.
>
> We are not sure how to correctly understand what "converse bounds" means. If the reviewer refers to lower bounds on the generalization error, we unfortunately do not have an immediate answer on how to address this -- in general, obtaining lower bounds on generalization error is an extremely nontrivial area of learning theory, and matching upper and lower bounds are exceedingly rare in the literature.

---

> ### Author Response · Authors · 2023-11-20
>
> Dear Reviewer X27N: Thank you again for your time and your feedback, which have helped us improve our paper. We hope our responses have adequately addressed your questions, and that you will reconsider your score in light of our discussion. As the discussion period ends soon, we would greatly appreciate it if you could let us know any remaining questions or concerns you may have: we are more than willing to provide additional explanations and experiments to clarify these.

---

> > ### Comment · Reviewer_X27N · 2023-11-22
> >
> > Thank you to the authors for their thoughtful responses. I am still concerned about the contributions of the paper, namely the strength of the theoretical results and the application of the bounds to actual practice. Based on that, I would like to keep my original rating. I would encourage the authors to think about focusing the paper on more specific results and how to apply those results in practice. Thank you.

---

### Author Response · Authors · 2023-11-16
**General Response to All Reviewers**

Dear Reviewers,

We thank you for the constructive feedback and for seeing the merits of our contributions. We have revised the paper to incorporate your suggestions and add empirical results: the revisions are colored in blue. We have also provided individual responses to each reviewer to address their comments. Below, we contextualize our work with respect to the relevant literature and clarify the key messages of our contributions. Please do not hesitate to reach out if there are any other questions and comments: we would be happy to discuss!

**Motivation.** As model architectures have become more and more complex (e.g., LLMs are parameterized with billions of parameters), their evaluation, training and fine-tuning become ever more challenging. Finding compressed architectures to reduce the number of trainable parameters has been growing in practical relevance (see the exploding literature on LoRA finetuning, for instance). Interestingly, a recent line of work argues that there is an interplay between compressible models and their ability to generalize to unseen test data. One compression scheme which has found success consists in training NNs on random, lower-dimensional subspaces. NNs compressed that way have been shown to yield satisfying test performances in various tasks while being faster to train (Li et al., 2018). **The primary goal of our work is to complement the understanding of NNs trained in this way, by deriving new information-theoretic bounds on their generalization error.**

**Impact of our contributions.** The overall unifying story of our paper is that random compression tightens information-theoretic generalization bounds. **This has two main consequences: 1. potentially informing practitioner's choices of NN architecture, and 2. bringing information-theoretic generalization bounds towards practical use.**
1) Our contributions provide a clear unambiguous message that NN architectures with low intrinsic dimension are likely to generalize well. Our bounds give a more nuanced portrayal of this phenomenon, as they explicitly depend on the subspace dimension $d$, number of quantization levels $L$, and number of training samples $n$. We believe that our findings can inform practitioners when choosing neural network architectures, e.g., we show that the generalization error can be controlled in practice by regularizing a *distortion term* (Section 4.2). Furthermore, we note that it may not even be necessary for practitioners to use compressed training: if a NN *without* compression has similar performance to its compressed version (i.e., the intrinsic dimension of the architecture is low), then **our theory provides a reason to believe that this NN architecture/design will tend to generalize well.** Empirically exploring this observation would be interesting future work, potentially leading to ranking NN architectures by intrinsic dimension and using this ranking to inform choice of architecture on new datasets.
2) The current weaknesses of information-theoretic generalization bounds are twofold: first, they scale in the number of trainable parameters and, as they ignore the existence of a lower intrinsic dimension, can be pessimistic; second, they require estimating a mutual information term, which has prohibitive computational cost in modern learning problems. For these reasons, information-theoretic bounds have, to our knowledge, not seen any practical use beyond very small toy examples. By considering NNs trained on random subspaces, we obtain generalization bounds that are tractable in regimes where the standard ones are fundamentally intractable, via the introduction of a mutual information term evaluated on the lower-dimensional subspace. Our work is thus a **significant step forward towards practical relevance of information-theoretic generalization bounds**. We refer to our reply to reviewer JYMx for a more technical discussion on how the theoretical guarantees of MI estimators, e.g. MINE, scale with $d$.

---

### Meta-Review · Area_Chair_mJiR · 2023-12-15

**Metareview:**

This paper analyzes mutual information generalization bounds for neural networks. It proposes a rate-distortion framework that allows for obtaining generalization bounds when the weights are close enough to the random subspace, and then it uses this result to develop a training procedure. The reviewers have found some values in this work but raised several concerns. The authors’ responses have answered some of the issues but failed to satisfactorily justify the strength of the theoretical results and the application of the bounds in practice. One reviewer questions the performance of the proposed models on more complex learning tasks and mentions that the theoretical contribution is somewhat limited.

**Justification For Why Not Higher Score:**

I reached this decision by evaluating the contributions and novelty of the work, taking into consideration both the reviews and the responses from the authors.

**Justification For Why Not Lower Score:**

N/A

---

### Decision · Program_Chairs · 2024-01-16

Reject